

# The metabolic response of thecosome pteropods from the North Atlantic and North Pacific Oceans to high $CO_2$ and low $O_2$

Amy E. Maas[1,2], Gareth L. Lawson[2] and Zhaohui Aleck Wang[3]

1. Bermuda Institute of Ocean Sciences, St. George's GE01, Bermuda

2. Biology Department, Woods Hole Oceanographic Institution, Woods Hole, MA, USA

3. Marine Chemistry & Geochemistry Department, Woods Hole Oceanographic Institution, Woods Hole, MA, USA

*Correspondence to*: Amy E. Maas (amy.maas@bios.edu)



**Abstract.** As anthropogenic activities, notably the burning of fossil fuels, increase carbon
dioxide ($CO_2$) and result in a decrease in oxygen ($O_2$) concentrations in the ocean system, it
becomes important to understand how different populations of marine animals will respond.
Water that is naturally lower in pH, with a high concentration of carbon dioxide (hypercapnia)
and a low concentration of oxygen, occurs at shallow depths (200-500 m) in the North Pacific
Ocean, whereas similar conditions are absent throughout the upper water column in the North
Atlantic. This contrasting hydrography provides a natural experiment to explore whether
differences in environment cause populations of cosmopolitan pelagic calcifiers, specifically the
aragonitic-shelled pteropods, to have a different physiological response when exposed to
hypercapnia and low $O_2$. Using closed-chamber end-point respiration experiments, eight species
of pteropods from the two ocean basins were exposed to high $CO_2$ (~800 µatm) while six species
were also exposed to moderately low $O_2$ (10%, or ~130 µmol kg$^{-1}$) and a combined treatment of
low $O_2$/high $CO_2$. None of the species tested showed a change in metabolic rate in response to
high $CO_2$ alone. Of those species tested for an effect of $O_2$, only *Limacina retroversa* from the
Atlantic showed a response to the combined treatment, resulting in a reduction in metabolic rate.
Our results suggest that pteropods have mechanisms for coping with short-term $CO_2$ exposure
and suggest that there can be interactive effects between stressors on the physiology of these
open ocean organisms that correlate with natural exposure to low $O_2$ and high $CO_2$; these are
considerations that should be taken into account in projections of organismal sensitivity to future
ocean conditions.








**Key Words: ocean acidification, zooplankton, respiration**





## 1. Introduction

Ocean acidification, a result of the dissolution of anthropogenically produced carbon dioxide ($CO_2$) into sea water, is increasingly considered to be one of the most pervasive human changes to the marine system (Doney et al., 2009; Gruber, 2011; Halpern et al., 2008). The pH of the ocean surface has already dropped by ~0.1 units relative to preindustrial levels and is predicted to drop another 0.3-0.4 pH units in the next one hundred years (Bopp et al., 2013; Haugan and Drange, 1996; IPCC, 2013). As $CO_2$ dissolves in the ocean, it causes changes in seawater carbonate chemistry, notably increasing hydrogen ion concentration and decreasing the concentration of carbonate ions. As a consequence of the changing equilibria, there is a reduction in pH and in the saturation state of calcium carbonate ($CaCO_3$), including the biogenic forms of calcite and aragonite. As ocean acidification continues, eventually the water becomes undersaturated and corrosive, meaning that, in the absence of compensating biological action, conditions will favor the dissolution of the $CaCO_3$ found in the shells and skeletons of calcifying organisms, with aragonite being more sensitive than calcite (Millero, 2007).

Ocean acidification, therefore, impacts calcifying species on multiple fronts. Changes in environmental pH can modify the acid-base balance of intra- and extracellular fluids of marine organisms, which may result in reduced fitness or outright mortality (Seibel and Fabry, 2003; Seibel and Walsh, 2001; Widdicombe and Spicer, 2008). Changes in $CaCO_3$ saturation state can also affect the ability of some calcifying animals to create and maintain calcium carbonate structures with implications for energetics, survival, competition and biogeochemical export (Fabry et al., 2008; Riebesell et al., 2000; Ries et al., 2009). Understanding the long-term effects of this increase in ocean acidity on both organisms and ecosystems has, therefore, become of great concern. Important and outstanding research goals are to understand how changing $CO_2$ impacts current populations and to predict whether these populations will be able to adapt to the rate and severity of the rising anthropogenic $CO_2$ inputs (e.g. Dam, 2013; Kelly and Hofmann, 2013; Sunday et al., 2011).

One approach to understanding the response of marine animals to increasing acidification is to examine places where animals already experience conditions of elevated $CO_2$ (hypercapnia). By comparing individuals that inhabit regions of high $CO_2$ with those that never experience high levels naturally, insight can be gained into the potential for adaptation of species to high $CO_2$ over evolutionary timescales. The ocean chemistry of the northwest Atlantic and the



northeast Pacific Oceans provides such a natural experiment. High $CO_2$ concentrations are generally absent from the upper water column in the Atlantic (Wanninkhof et al., 2010). In contrast there currently are hypercapnic conditions, where the water is undersaturated with respect to aragonite, in the upper water column in parts of the Pacific.

The source of hypercapnia in the Pacific Ocean is a combined result of ocean circulation coupled with the biological processes, leading the old deep waters of the Pacific to be some of the most $CO_2$ rich in the ocean (Broecker et al., 1982). On top of this natural process, ocean acidification also plays a role: the pH of the upper water column in the North Pacific is decreasing by on the order of 0.001–0.002 pH units per year (Byrne et al., 2010), corresponding to a total $CO_2$, or dissolved inorganic carbon (DIC), increase of 1–2 µmol $kg^{-1}$ $yr^{-1}$ (Peng et al., 2003; Sabine et al., 2008; Sabine and Tanhua, 2010). Although the surface waters in these regions are typically well oxygenated and with a pH > 8, animals that live at or migrate to depth experience increasingly low oxygen ($O_2$), pH, undersaturation with respect to calcium carbonate and elevated $CO_2$ (Seibel, 2011). Historically these regions, which occur in many ocean basins, were in fact known more for their low $O_2$ than for their high $CO_2$ and were termed oxygen minimum zones (OMZs). These carbon maximum/oxygen minimum zones are extensive in the North Pacific Ocean, whereas similar conditions are rare in much of the Atlantic (Paulmier et al., 2011). Closely related taxa and cosmopolitan species in these two regions therefore experience very different pH levels, $CO_2$ and $O_2$ concentrations in their normal distribution. Independent from high $CO_2$, the reduced $O_2$ at depth in these OMZs has a profound impact on zooplankton distribution (i.e.: Escribano et al., 2009; Maas et al., 2014; Wishner et al., 2008) and can have important implications for the physiology of zooplankton (Childress and Seibel, 1998; Rosa and Seibel, 2008; Seibel, 2011).

Thecosome pteropods are an interesting group for investigating planktonic exposure and response to hypercapnia and low $O_2$. Broadly distributed throughout the open ocean, species of thecosomes found in shallow waters of temperate and polar seas can become a numerically dominant member of the zooplankton community (Bednaršek et al., 2012a; Hunt et al., 2008; van der Spoel, 1967). As such, they can be an important part of the food chain (Armstrong et al., 2005; Hunt et al., 2008; Karnovsky et al., 2008), and contribute substantially to carbon flux (Bauerfeind et al., 2009; Fabry and Deuser, 1991; Manno et al., 2010; Noji et al., 1997). Bearing thin shells of aragonite, one of the less stable forms of biogenic calcium carbonate, the



calcification of thecosomes has been shown to be impacted by exposure to conditions replicating
the projected changes in surface water pH and saturation state of the future ocean in the next 100
years (Comeau et al., 2009; Lischka et al., 2011; Manno et al., 2012). Furthermore, recent
assessments have shown that their shells already bear the mark of acidification in upwelling and
polar regions characterized by under-saturated conditions with respect to aragonite (Bednaršek
et al., 2014a, b; Bednarsek and Ohman, 2015; Bednaršek et al., 2012b). Studies of metabolism
and behavior, however, reveal a complex sensitivity to pH, dependent upon natural pre-exposure
and the presence of interactive stressors (Comeau et al., 2010a; Maas et al., 2012a; Manno et al.,
2012; Seibel et al., 2012).

Previous work has shown that some tropical and sub-tropical thecosome species undergo

diel vertical migrations into persistent and pronounced regions of low $O_2$ and hypercapnia in the
Eastern tropical North Pacific. These species showed no change in metabolic rate ($O_2$
consumption) when exposed to high $CO_2$ (1000 µatm), revealing the ability of some groups of
thecosome to maintain aerobic metabolism in acidified waters for short periods of time. The one
species in this region that does not migrate, however, responded with a suppression of
metabolism when exposed to high $CO_2$ (Maas et al., 2012a). This work in the Eastern tropical
North Pacific provides evidence that there may be the potential for environmental adaptation of
thecosomes to high $CO_2$, but provides no insight into the combined effects of $CO_2$ with low $O_2$.
Although research into this topic is underway for other calcifying organisms in coastal habitats
(Gobler et al., 2014; Melzner et al., 2013), in the open ocean our understanding remains limited.

The objective of this study, therefore, was to compare the effect of high $CO_2$ and low $O_2$

on thecosome pteropods from the northwest Atlantic and the northeast Pacific Oceans. One of
the benefits of this comparison is that there are a number of species of thecosomes that have
cosmopolitan distributions occupying both basins and that are known to be diel vertical
migrators (Table 1; Bé and Gilmer, 1977; van der Spoel, 1967). Thus populations in the Pacific
would naturally experience hypercapnia and low $O_2$ in their daytime deep water habitat in the
Pacific, while in contrast, those from the Atlantic would rarely experience the same deep water
environmental stressors. Using these organisms, which are presumably adapted to their local
conditions, we can test whether species exhibit a population-specific physiological response to
these environmental conditions indicative of different sensitivities.



**2. Methods**
Thecosome pteropods caught during cruises to the northwest Atlantic and northeast Pacific were
exposed aboard ship to manipulated conditions of moderately high $CO_2$ and/or low $O_2$ for short
durations (< 18 h). After this exposure their metabolic rates were measured and then compared to
determine whether there were species- or region-specific responses to the treatments.
**2.1 Sampling**
Animals were collected on two cruises, the first on August $7^{th}$ – September $1^{st}$ 2011 in the
northwest Atlantic aboard the R/V *Oceanus*, and the second in the northeast Pacific from August
$9^{th}$ – September $18^{th}$ 2012 aboard the R/V *New Horizon*. The majority of the sampling in the
Atlantic took place along a three-part 'z'-shaped transect running between 35°N 52°W and 50°N
42°W, as well as at sites during transit to and from port (Fig. 1). The first portion of this cruise
track corresponded to a segment of the World Ocean Circulation Experiment / Climate and
Ocean: Variability, Predictability and Change project (WOCE/CLIVAR) line A20. In the North
Pacific the main sampling took place along a two-part transect running between 50°N 150°W
and 33.5°N 135°W, corresponding to a portion of WOCE/CLIVAR line P17N, as well as at sites
during transit to and from port (Fig. 1).
Sampling was part of a larger interdisciplinary project employing a suite of tools to
explore the natural distribution and hydrographic environment of the thecosomes. The sampling
design included underway measurements of hydrography, carbonate chemistry and multi-
frequency acoustic backscatter. Comprehensive sampling of the water column was conducted at
pre-determined stations using a depth-stratified 1-$m^2$ Multiple Opening/Closing Net and
Environmental Sensing System with 150 µm mesh nets (MOCNESS; Wiebe et al., 1985), a
towed broadband echosounder, video plankton recorder casts, and profiles with a 24 10-L Niskin
bottle rosette and associated conductivity, temperature and depth (CTD) package.
Hydrographic profiles associated with this study were collected of temperature, $O_2$ and
salinity using the CTD-Rosette-Niskin bottle package at stations along the main survey transects
(Fig. 1). This CTD was equipped with dual temperature and conductivity sensors, a Digiquartz
pressure sensor, a SBE43 dissolved oxygen sensor, a biospherical underwater photosynthetically
active radiation (PAR) sensor with surface reference, a Wet Labs C-Star transmissometer (660
nm wavelength), and a Wet Labs ECO-AFL fluorometer. Where CTD casts were unavailable, at
stations conducted during the transits to and from port, an expendable bathythermograph (XBT)



was deployed to determine the temperature of the water column. Bottle samples of carbonate
parameters, nutrients, and other parameters were collected at selected water depths using the
CTD-Rosette package.
**2.2 Environmental Carbonate Chemistry**
Discrete pH samples were directly collected from the 10-L Niskin bottle into 10 cm cylindrical
optical cells and measured within 4 h of collection (Clayton and Byrne, 1993; Dickson et al.,
2007). These pH samples were analyzed spectrophotometrically on an Agilent 8453
spectrophotometer at a control temperature (25.0 ± 0.1°C) following the method detailed in
Dickson (2007) and in Clayton and Byrne (1993) using m-cresol purple as the indicator. The pH
results in total scale have been corrected for indicator impurity (Liu et al., 2011) and indicator
perturbation to seawater samples. The pH measurements have a precision better than 0.001 and
an accuracy of ~0.002.
Nutrient samples (nitrate/nitrite, phosphate, silicate, and ammonia) were collected in 20
mL plastic bottles after filtration through a 0.22um Pall capsule filter and kept frozen until
analysis. Nutrient samples were analyzed either at the WHOI Nutrient Analytical Facility or the
University of California, Santa Barbara, using a Lachat Instruments QuickChem 8000 four-
channel continuous flow injection system, following standard colorimetric methods approved by
U.S. Environmental Protection Agency.
Discrete samples were also taken for dissolved inorganic carbon (DIC) and total
alkalinity (TA). These were collected in 250mL Pyrex borosilicate glass bottles after being
filtered with a 0.45 µm in-line capsule filter and poisoned with saturated mercuric chloride
(Dickson et al., 2007). DIC samples were analyzed on a DIC auto-analyzer (AS-C3, Apollo
SciTech, Bogart, USA) via sample acidification, followed by non-dispersive infrared $CO_2$
detection (LiCOR 7000: Wang et al., 2013; Wang and Cai, 2004). The instrument was calibrated
with certified reference material (CRM) from Dr. A.G. Dickson at the Scripps Institution of
Oceanography. The DIC measurements have a precision and accuracy of ±2.0 µmol kg$^{-1}$. TA
measurements were made with an Apollo SciTech alkalinity auto-titrator, a Ross combination
pH electrode, and a pH meter (ORION 3 Star) based on a modified Gran titration method with a
precision and accuracy of ±2.0 µmol kg$^{-1}$ (Wang and Cai, 2004).
The remaining water column carbonate system parameters, including aragonite saturation
state and $pCO_2$ were calculated from DIC-pH pairs at in situ nutrient, temperature, salinity and



pressure using the software CO2Sys (Pierrot et al., 2006) and the dissociation constants of
Mehrbach et al. (1973), refitted by Dickson and Millero (1987), and the $KHSO_4$ dissociation
constant from Dickson (1990). Depths for pH=7.7, $pCO_2$=800 µatm and aragonite saturation
state of 1 were then linearly interpolated using the closest available measurements.

Surface water $pCO_2$ was continuously measured throughout both cruises using an

automated underway system (Model 8050, General Oceanics Inc., USA) based on headspace air-
seawater equilibration followed by infrared detection (LiCOR 7000). This system was calibrated
every 1-2 hours with three $CO_2$ gas standards traceable to World Meteorological Organization
$CO_2$ Mole Fraction Scale. These underway $pCO_2$ measurements have a precision and accuracy of
~±1 µatm. Measurements made by the underway system provide insight into the carbonate
chemistry parameters at stations made in transit where bottle samples were not collected.

**2.3 Specimen Capture**

Thecosome species were chosen for physiological study opportunistically as they appeared in net
samples at successive stations. Species were targeted specifically for their abundance and the
likelihood of their presence in both ocean basins. Most individuals were collected with a 1-m
diameter, 150-µm mesh Reeve net with a ~25 L cod-end in the Atlantic and a similar 1-m
diameter, Reeve net equipped with 330-µm mesh in the Pacific. Use of the Reeve net with its
large and heavy cod-end in combination with slow haul rates (typically 5-10 m min$^{-1}$) allowed
for a gentle collection of the delicate thecosomes, consistently supplying animals in good
condition with undamaged shells and external mantle appendages. Net tows were made at night
when animals were expected to congregate at shallow depths, were ~1 h in duration in an effort
to minimize the handling time of the organisms, and reached a maximal depth between 100–150
m. Depths were targeted that had a high chlorophyll $a$ peak during CTD casts, high acoustic
backscattering on the echosounder, and/or where thecosomes had been abundantly sampled at
the same station using the MOCNESS. Occasionally, individuals of less abundant species were
collected from the nets of the MOCNESS for physiological study, but only if their shells were
undamaged and they were swimming normally.

Post-capture, individuals were transferred to filtered water in densities of < 15 ind. L$^{-1}$

and kept for at least 8 h in temperature controlled waterbaths to allow for gut clearance.
Temperatures for experimentation (20, 15 or 10°C) were chosen to be generally representative of
the waters from which the animals were sampled, based on the vertical distributions and



hydrographic conditions documented with the stratified MOCNESS sampling. Chosen
temperatures were typically the average of the water temperature between 25-100 m, although in
the middle section of the Atlantic cruise experimental temperatures were reflective of the 25–50
m average due to the particularly shallow vertical distribution of the dominant species (*Limacina*
*retroversa*) sampled in this region. This was to ensure that experiments were occurring at
physiologically relevant and, presumably, natural temperatures for each species. After gut
clearance, individuals that were in good condition (i.e., swimming and with shell intact) were
used for oxygen consumption experiments.

**2.4 Experimental Exposures and Oxygen Consumption Rate**

Post-gut-clearance, healthy individuals were put into separate glass syringe respiration chambers
with a known volume of 0.2 µm filtered seawater and 25 mg $L^{-1}$ each of streptomycin and
ampicillin. This minimized the microbial respiration effects on the measurements of carbonate
chemistry and $O_2$ consumption rates by pteropods during the experiments. The inclusion of
antibiotics, a method which has previously been used with thecosomes to prevent bacterial
growth in respiration experiments (Maas et al., 2012b), was shown during the Pacific cruise to
have no effect on the $O_2$ consumption of at least *Limacina helicina*, for the exposure durations
associated with these experiments (Howes et al., 2014). The volume of water in the treatments
was chosen to complement the size of the organism and temperature of the experiment and
ranged between 15-50 mL in 2011 and 8-20 mL in 2012. For every 3-5 treatment chambers, a
"control" respiration chamber (experimental seawater with antibiotics and without pteropods)
was set up to monitor microbial activity and to provide water for characterization of the starting
conditions.

Filtered seawater for experimental exposures was collected during both cruises in batches

at approximately weekly intervals from the surface; experimental water thus began with
chemical properties (notably including TA, DIC, pH, as well as salinity) reflective of the local
environment and was then manipulated to modify $CO_2$ and/or $O_2$ concentrations. Manipulations
were achieved by bubbling 1 L batches of collected seawater with gas mixes (certified accurate
to ± 2%) for 45–60 min with one of two oxygen (21 and 10% $O_2$) levels crossed with two $CO_2$
(nominally 380 ppm and 800 ppm) levels. At the time of the experiment, surface air $pCO_2$
conditions were on average ca. 380 ppm, dictating our ambient (LC) conditions. In 2011 the



ambient condition (~21% $O_2$ and 380 µatm $CO_2$) was achieved by bubbling with an ambient
clean air line, while in 2012 it was achieved by a certified 380 ppm gas mix.

The experimentally modified concentrations mimic the $CO_2$ and $O_2$ levels that would be

experienced by thecosomes within the top 400 m of the Pacific Ocean, and reflect the average
projected atmospheric $CO_2$ level for the open ocean in the year 2100 (A2 emissions scenario,
IPCC, 2007). This resulted in four total treatments: low (i.e., ambient) $CO_2$, high oxygen
(LC/HO) representative of current ambient surface ocean conditions; high carbon, high oxygen
(HC/HO), replicating what we expect average future surface oceans to resemble; low $CO_2$, low
oxygen (LC/LO); and high carbon, low oxygen (HC/LO), which is similar to what organisms in
the Pacific would experience during a diel vertical migration into the local oxygen minimum
zone. The goal of this design was to allow us to compare directly the Atlantic and Pacific
thecosomes to see whether exposure to 800 µatm $pCO_2$ and/or 10% $O_2$ resulted in different
outcomes. The level of low $O_2$ chosen for this study was well above the threshold that has been
designated as stressful for non-specialized metazoan life (< 2 mg $O_2$ $L^{-1}$ or 60 µmol $O_2$ $kg^{-1}$;
Vaquer-Sunyer and Duarte, 2008), in order to test the non-lethal effect of moderately low $O_2$ on
individuals from the two ocean basins. Calculations based on the salinity and temperature of the
water indicated that bubbling with 10% $O_2$ achieved conditions of 10–13% $O_2$ saturation at the
start of experiments. Subsequent analyses (see below) also confirmed that intended $CO_2$
concentrations were achieved for all treatments within reasonable ranges, with the exception of
the LC/LO Atlantic treatment. In this case, the gas cylinder was evidently improperly mixed by
the manufacturer and analyses suggested a ca. 100 ppm $CO_2$ concentration. The results for this
treatment are still presented but should be interpreted as a distinct treatment.

Oxygen consumption was measured following similar techniques as described in Marsh

and Manahan (1999). Briefly, at the conclusion of the experiment water was withdrawn from
treatment or control chambers using an airtight 500 µL Hamilton syringe and injected past a
Clarke-type microcathode (part #1302, Strathkelvin Instruments, North Lanarkshire, United
Kingdom) attached to an $O_2$ meter (part #782) in a water-jacketed injection port (part #MC100).
This was done three times, allowing the reading to stabilize for at least 30 seconds before a
measurement was taken. Generally, the change in oxygen consumption was between 3–25% of
the control value. In high oxygen experiments, if the oxygen level fell below 70% of air
saturation they were excluded from the analysis. Animals were removed from the chamber,



blotted dry and frozen in liquid nitrogen. These individuals were later weighed using a
microbalance ($\pm$ 0.0001 g) and the resulting mass specific $O_2$ consumption rates are reported in
$\mu$moles (g wet weight)$^{-1}$ h$^{-1}$. Wet weights are here used as they are more relevant for
physiological understanding of animal function (Childress et al., 2008) but dry weights can be
estimated from these using the wet weight to dry weight relationships developed previously for
pteropods (Ikeda, 2014). To replicate the duration of exposure that would be experienced by
most thecosomes in the Pacific undergoing a daily migration to depth, the experiments were
targeted to last 6–12 h. In practice, experiments ranged from 6–18 h for normoxic and 3–10 h for
low $O_2$ trials. This variation in duration resulted from balancing the need to elicit a measureable
change in $O_2$ concentration with preventing extreme $O_2$ depletion of the chambers (< 6% oxygen
saturation) and accounting for multiple species of variable size and metabolic rate.

**2.5 Experimental Carbonate Chemistry**

Carbonate chemistry of the treatments was characterized in most cases via measurements of DIC
and TA of experimental seawater, unless indicated otherwise. The process of measuring the $O_2$ in
the treatments used up a large portion of the water and then the chamber was unsealed and
disturbed to remove the animal, rendering it impractical to measure the carbonate chemistry
directly from the respiration chambers. DIC measurements were thus taken from control syringes
within 18 h of the end of each experiment and used to represent the starting point of the
carbonate chemistry conditions the animals experienced. Water samples were allowed to come to
room temperature (> 6 h) before analysis. DIC was measured using the same system as that used
for the hydrographic characterization (see above). Estimates of the effect of $CO_2$ production via
respiration in treatment chambers on DIC were made using a respiratory quotient of 0.8 M of
$CO_2$ per 1 M of $O_2$ consumed (Mayzaud, 1976) to characterize the ending conditions of the
experiments.

Due to the small volumes of water in the experimental chambers, it was not possible to

measure both DIC and TA from the control syringes. Instead, TA samples intended to be
representative of the starting experimental conditions were collected via siphoning from each
batch of filtered and antibiotic-treated water. These samples were subsequently measured based
on the analytical method described above (Wang and Cai 2004). TA of experimental water was
assumed to have been constant over the course of each experiment as water was filtered (0.2 $\mu$m)



and antibiotic treated (thus microbial activities were kept at minimum), and aerobic respiration
does not change TA in a significant way.
In some instances, however, measured TA from experimental water was substantially
dissimilar to that of the surface measurements made from nearby in-situ surface bottle samples
collected with the CTD (> 20 $\mu$mol kg$^{-1}$; see section 3.3). Calculated $pCO_2$ values in these cases
were also significantly different from batches of experimental water collected from other
locations, but bubbled with the same $CO_2$ gas tank. These differences are more than 10 times the
measurement precision/accuracy and 5 times the uncertainty of duplicate sampling and
measurements during the cruises. They are also beyond the likely level of TA variation due to
differences in sampling location (geographic and in depth) between the in situ bottle samples and
experimental water batches and rather are likely a consequence of the difficulties associated with
cleanly siphoning the experimental water batches (e.g., contamination during sampling). For
completeness, the carbonate chemistry system parameters for the experimental water, including
aragonite saturation state and $pCO_2$, are reported based on calculations using DIC-TA pairs using
both the in situ and experimental TA; in those cases where the TA measurements diverged
substantially (> 20 $\mu$mol kg$^{-1}$), however, we base our interpretations on the in-situ measured TA
at nearby CTD stations instead of the values of experimental water. In those circumstances
where batch water was taken from test stations and CTD bottle data were unavailable, the
experimental TA was checked using calculated TA values using DIC from the LC/HO treatments
and $pCO_2$ from the underway measurements.
**2.6 Statistics**
Oxygen consumption rates were tested for significant differences between groups with
Bonferroni pairwise post-hoc comparisons using SPSS. Univariate General Linear Models
(GLM) were conducted to determine the effect of $CO_2$ level, $O_2$ level, and their interactive effect
using the log transformed oxygen consumption with log transformed wet mass as a covariate
separately for each species (2 factor design; "$CO_2 \times O_2$"). In the Atlantic this full factorial design
was confounded by the incorrect gas mixture so each treatment was tested independently (1
factor design; "treatment"). Species that were collected during both years/basins, and
experiments conducted on species at multiple temperatures, were analyzed separately so that the
effect of variations in mass between seasons and the changes in metabolic rate at different
temperatures would not confound the analysis.




For some species the temperature of experimentation was different among stations within
a basin. For analyses with these species when comparing species between ocean basins, we
applied a standard temperature coefficient ($Q_{10}$) to compare across temperatures. The adjusted
rates ($R_f$) were calculated at 15°C using a $Q_{10}$ of 2 according to the equation:
$$R_f = R_i * \left( Q_{10}^{\left( \frac{15 - T_i}{10} \right)} \right)$$

where $R_i$ is the original metabolic rate measured at the original temperature ($T_i$). Although
previous work with thecosomes has shown that $Q_{10}$ is species-specific (Maas et al., 2011; Maas
et al., 2012b; Seibel et al., 2007), for many of the species used in this study there are no
published estimates of $Q_{10}$. Thus, this coefficient value was chosen as it is mid-range for the
published $Q_{10}$ of non-polar thecosome species as recently compiled by Ikeda (2014; 1.3-2.7) and
is consistent with estimates of average $Q_{10}$ for marine ectotherms, which typically fall between
2-3 (Hochachka and Somero, 2002; Seibel and Drazen, 2007).

**3.  Results**
**3.1 Specimen Capture**
Following currently accepted taxonomy, individuals from a total of eight species were collected
over the course of the two cruises for physiological studies. The taxonomy of thecosomes has
recently begun to be revisited using molecular and paleontological tools (i.e. Hunt et al., 2010;
Janssen, 2012; Jennings et al., 2010; Maas et al., 2013), however, and there is growing evidence
of cryptic speciation for some pteropod groups (Burridge et al., 2015; Gasca and Janssen, 2014).
It thus should be noted that these inter-basin comparisons may be of cryptic congeners rather
than conspecific populations.
We collected two species of thecosome pteropods exclusively from the Atlantic,
*Limacina retroversa* (Fleming, 1823), a subpolar species, which is absent from the North Pacific,
and *Diacria trispinosa* (Blainville, 1821), which can be found in temperate and tropical regions
of the Atlantic, Pacific and Indian Oceans. Although present in both the North Atlantic and
Pacific, the polar to sub-polar species *Limacina helicina* (Phipps, 1774), was only sampled in the
Pacific transect. Collections of this species consisted of intermixed formae, the high spiraled
*Limacina helicina helicina acuta* (van der Spoel, 1967), the lower spiraled *Limacina helicina*
*helicina pacifica* (van der Spoel, 1967), and a forma that bore resemblance to both in a mixed



morphology. Since both the assemblage and morphology of these formae were mixed they were
tested as one population/species. In both ocean basins we collected *Styliola subula* (Quoy and
Gaimard, 1827)*, Cavolinia inflexa* (Lesueur, 1813) and *Clio pyramidata* (Linnaeus, 1767).
There is some morphological and molecular evidence that *Cuvierina columnella* (Rang, 1827) is
actually multiple distinct species, now including *Cuvierina atlantica* and *Cuvierina pacifica*
(Burridge et al., 2015; Janssen, 2005), and we tested individuals of these species from their
respective ocean basins.
**3.2 Hydrography**
Two hydrographic regimes were evident along the North Pacific study transect (Table 2; Fig. 2).
The northern-most stations, including portions of the transit from port and stations from 50°N
150°W to 47 °N 144.6°W were coldest, with the temperatures between 25-100 m ranging from
5-10°C. In this area $O_2$ fell below 10% (~130 µmol kg$^{-1}$) by 250 m. In this northern part of the
transect, pH fell below 7.7 by 130 m, and $pCO_2$ had already reached 800 µatm by ~200 m.
Individuals in this area experienced an $\Omega_{Ar}$ = 1 between 160-185 m, well within the typical diel
vertically migratory range of both of the species found in the region (*C. pyramidata* and *L.*
*helicina*). At stations from more southern latitudes, from 47 °N 144.6°W to 33.5°N 135°W,
temperatures at depths between 25-100 m were higher, ranging between 10-15°C, representative
of the transition zone into the North Pacific Gyre. Along this portion of the transect $O_2$
concentration consistently fell below 10% by depths between 340 and 400 m. The depth at which
pH fell below 7.7 increased gradually from ~150 to 230 m as latitude decreased. Similarly, the
depth at which $pCO_2$ in this area reached 800 µatm deepened from 330 to 440 m, and the
aragonite saturation horizon transitioned from 330 m to 430 m depth. The depth at which species
would experience a pH below 7.7 was within the inhabited depth range known from the literature
for all of the species tested in this study region, but only the species *Clio pyramidata* likely
experienced 10% $O_2$, 800 µatm $pCO_2$ and aragonite undersaturation in its typical distribution in
this portion of the Pacific transect (Table 1).

In contrast to the Pacific, along the entire Atlantic transect $O_2$ concentration was above

~200 µmol kg$^{-1}$ in the top 500 m, while $pCO_2$ never reached 800 µatm and aragonite
undersaturation never occurred throughout the top 1000 m. There were three dominant
hydrographic regimes in the Atlantic (Table 2; Fig. 2). In the northeastern part of the sampling
region (50°N 42°W to 44.9 °N 42°W), where the Gulf Stream meets the Labrador Current,





average temperatures at 25-100 m were near 15°C and pH only fell below 7.7 at depths
exceeding 400 m. Similarly, in the southwest part of the sampling region (from 42°N 52°W to
36°N 52°W), corresponding to the Sargasso Sea and through the Gulf Stream, pH only fell below
7.7 at depths exceeding 450 m, although the upper water column was warmer, with average
temperatures being 20°C. There was a third water mass type, typical of colder fresher shelf
waters, at station 32 and in an intrusion off the Grand Banks at stations 17 and 19. This water
was typified by a temperature and salinity anomaly with average temperatures falling below 5°C
from 25-100 m and a salinity signature < 33, contrasting significantly with the surface salinities
of the northern portion (~34) and southern portion (~36) of the Atlantic transect. As a
consequence, these stations contained water of the lowest pH, with surface waters reaching 7.7 at
depths shallower than 200 m. Based on previous knowledge of the vertical distributions of the
thecosomes used in this study, only the species *Clio pyramidata* would ever experience a pH
below 7.7 in this overall Atlantic study region and none of the thecosomes studied would
experience 800 µatm $pCO_2$ or under-saturation within their vertical range (Table 1).
**3.3 Carbonate Chemistry of Experiments**
Bubbling with $CO_2$ levels of ~380 and ~800 ppm resulted in a distinct separation of carbonate
chemistry between treatments during the experiments in both oceans (Table 3). Due to pre-
existing differences in the carbonate chemistry of the seawater collected in each ocean, TA
concentrations were different between the two basin treatments. In the Atlantic the DIC of the
ambient $CO_2$ treatments ranged from 2030-2090 µmol kg$^{-1}$ and the high $CO_2$ treatments from
2140-2220 µmol kg$^{-1}$, with an average difference between treatments of similar temperature and
salinity of 132 µmol kg$^{-1}$. Surface TA in the region decreased from ~2370 µmol kg$^{-1}$ in the
southern part of the transect to 2300 µmol kg$^{-1}$ in the northern latitudes. In the Pacific the DIC of
the ambient $CO_2$ treatment ranged from 1930-2020 µmol kg$^{-1}$ and the high $CO_2$ treatment from
2030-2110 µmol kg$^{-1}$, with an average difference of 90.7 µmol kg$^{-1}$ between the treatments.
Surface TA in this basin was 2150 µmol kg$^{-1}$ in the most northern collection and had decreased
to 2200 µmol kg$^{-1}$ by the transect mid-point.
Calculations of $pCO_2$ based on these measurements of DIC and TA suggested that target
$pCO_2$ levels were generally attained and were consistent between the two cruises, with the
exception of the LC/LO treatment in the Atlantic. In this case, there was a substantial deviation
from the intended $pCO_2$, suggesting values ranging from 99-139 µatm in contrast to a range of





311-391 µatm for the LC/HO in the Atlantic and 283-409 µatm for LC/HO and 295-397 µatm in
the LC/LO in the Pacific. Evidently, this indicates improper mixing of the gas concentration in
the Atlantic LC/LO gas cylinder by the manufacturer. The calculations for the high $CO_2$
treatments were more consistent between cruises, with the HC/HO being 585-868 µatm for $pCO_2$
and the HC/LO being 755-783 in the Atlantic, while in the Pacific the HC/HO treatment was
between 520-740 µatm and the HC/LO 546-766 µatm. The variability in calculated $pCO_2$ values
likely represents variations in bubbling time, temperature, and the degree to which the water
reached saturation relative to the gas mixtures. The variability within each distinct treatment may
also reflect, to some degree, what pteropods may experience under that particular mean
condition, i.e. low vs. high $CO_2$.

As a consequence of the natural differences in seawater carbonate chemistry, in particular

the TA differences between two ocean basins, there were inherent differences in the aragonite
saturation state between the Pacific and Atlantic treatments (Table 3). In the Atlantic $\Omega_{Ar}$ of the
ambient $CO_2$ treatment ranged from 2.4-3.5, except for the LC/LO treatment ($\Omega_{Ar}$ 4.0-5.5), which
was bubbled with an incorrect gas mixtures as discussed above. Comparatively, in the Pacific the
ambient $CO_2$ condition had a lower range of $\Omega_{Ar}$ (2.2-2.4) for both the LC/HO and the LC/LO
treatments. The experimental conditions of the high $CO_2$ treatments in the Atlantic only
approached under-saturation in the middle part of the transect ($\Omega_{Ar}$ = 1.2 at mid-latitudes; Table
3), where cold northern waters of low salinity were encountered and $\Omega_{Ar}$ had a range of 1.5-2.0
for the rest of the transect in the Atlantic. The values of $\Omega_{Ar}$ were lower overall in the Pacific,
although the high $CO_2$ treatments also never reached under-saturation ($\Omega_{Ar}$ 1.3-1.8).  The
manipulation of carbonate chemistry in general successfully created two distinct ranges for both
$pCO_2$ and aragonite saturation state ($\Omega_{Ar}$) in this study.

It is important to acknowledge that the production of $CO_2$ via respiration of the organisms

within the chambers would modify the carbonate chemistry of the treatments over the duration of
the experiments. Based on the average respiration rate, and using a respiratory quotient of 0.8
(Mayzaud, 1976), we estimate an average DIC production of ~18.0 µmol kg$^{-1}$ by the end of an
experiment. Applying such a change to the experimental conditions in the northeast Pacific,
where seawater is more sensitive to changes in DIC due to a lower buffering capacity compared
to the Atlantic (i.e., a worst case scenario), $\Omega_{Ar}$ would only change by <0.1 in both the LC and
HC experimental chambers over the course of the respiration experiments. Although this is an





appreciable effect, we nonetheless retain a wide separation between the ambient and high $CO_2$
treatments and in no cases would the treatments reach under-saturation as a consequence of this
biological activity. As such, for simplicity the results reported in Table 3 do not include this
variability.

### 3.4 Oxygen Consumption Rate

#### 3.4.1 Effect of $CO_2$

Varying availability and abundances of the different thecosome pteropod species in the net
samples precluded all species being exposed to the full factorial design but individuals of all
species were tested under the low $CO_2$, high oxygen (LC/HO) and high carbon, high oxygen
(HC/HO) treatments (Fig. 3, Table 4). To explore differences in metabolism attributable to a
response to $CO_2$, the log transformed wet mass was used in a GLM as a covariate comparing the
log transformed oxygen consumption (response variable) under low and high $CO_2$ conditions;
each population within a species that was sampled in both basins or run at multiple experimental
temperatures, was examined separately. There was no significant effect of $CO_2$ for any species in
either basin.

#### 3.4.2 Effect of basin

Following this assessment, we were interested in determining whether there were
between basin differences in metabolic rate. As such we ran a GLM using log transformed
metabolic rates for the three species that were found in both basins, normalized to 15 °C to
account for differences in experimental temperature by applying a standard temperature
coefficient. With the log-transformed wet mass as a covariate, we tested for an effect of basin,
$CO_2$ and an interactive term. *Clio pyramidata* had a similar metabolic rate between basins. In
contrast, *Cavolinia inflexa* ($F_{1,20}$=10.358, p=0.004) and *Styliola subula* ($F_{1,23}$=11.817, p=0.002)
both had a significantly lower metabolic rate in the Pacific, although no interactive effect of $CO_2$.

#### 3.4.2 Effect of $O_2$

For the species where enough individuals were collected to provide experimental
replicates to explore the interactive effects of $CO_2$ and $O_2$ we also ran a species and basin
specific GLM exploring the effect of treatment (Fig. 3, Table 5). *Clio pyramidata*, the only
species we were able to test in both basins showed no significant effect of high $CO_2$, low $O_2$ or
the interactive treatment in either basin. In the Pacific, *L. helicina* and *C. inflexa* similarly
showed no significant change in metabolic rate as a consequence of any of the treatments.  In





contrast, in the Atlantic, there was a significant effect of treatment for *L. retroversa* and a
Bonferroni post-hoc analysis comparing the treatments found that the high $CO_2$, low $O_2$ (HC/LO)
treatment was significantly lower than all other treatments (Fig. 4A; $F_{3,38}$=17.836, p<0.001; a
~60% reduction in the average mass specific metabolic rate in comparison with the LC/HO
treatment; Table 4). *Cuvierina atlantica* was tested at both 15 and 20 °C in the Atlantic, so to
make comparisons among these experiments a temperature coefficient was applied and rates
were normalized to 15 °C, after which no significant effect of any treatment was found for this
species.

**4. Discussion**
This study reveals that short term exposure to low $O_2$ and high $CO_2$, similar to what would be
experienced by individuals in the Pacific during diel vertical migration, does not influence the
oxygen consumption of most of the thecosome pteropod species examined from either the
Atlantic or Pacific. The only species which had a significant change in respiration in response to
any of the treatments was *Limacina retroversa* from the Atlantic, which responded to the
combined effect of low $O_2$ and high $CO_2$ with a reduction in oxygen consumption rate.
**4.1 Experimental Design**
A factor that should be considered when interpreting our results is the dynamic hydrographic
conditions that the animals experience naturally between and within the ocean basins.
Thecosomes of multiple species were found at a range of temperatures, salinities and carbonate
chemistries, meaning that they experienced a range of pH and aragonite saturation states in their
natural habitat. When comparing animals from multiple locations, we chose to use local water in
order to replicate these natural conditions and to manipulate exclusively the $CO_2$ concentration,
as this is the factor that is changing due to anthropogenic activity. This approach, however, does
not control for the other parameters of the carbonate chemistry system, which will vary between
regions. Despite this fact, there was a clean distinction between treatments, notably in terms of
aragonite saturation state as well as $CO_2$ concentration, that provides insight into the effect of
moderate short duration exposure to $CO_2$.

It is also important to note that the individuals of *L. helicina* from the Pacific experiments

did occasionally have very high mortality during the period prior to experimentation (>80% at
transit station T2 and T5, decreasing substantially to the northwest and along the main Pacific



transect). These individuals, which are polar/sub-polar organisms and are typically found
between -2 to 10 °C (Lalli and Gilmer, 1989), were collected from water that was likely near the
upper limit of their optimal temperatures. Animals collected from these sites that were used in
subsequent respiration experiments may therefore have been taken from an already stressed
population of individuals and should be recognized as such.
**4.2 Carbon Dioxide Effect**
Hydrographic profiles collected in the Pacific coincident to sampling of thecosomes, indicate
that organisms in the northern portion of the study region would experience conditions of high
$CO_2$ and low $O_2$ in the upper ~450 m of their distribution (Chu et al., in review), unlike in the
Atlantic. Despite these environmental differences, we found no significant effect of increasing
$CO_2$ alone on the respiration rates of any of the species from either ocean basin. These results
increase the published evidence that short term (6-18 h) exposure to enhanced $CO_2$ without
synergistic stressors has no significant effect on the metabolic rate of many species of thecosome
pteropods. Thus far, there are only two species that have been documented to show a change in
metabolism based on exposure to manipulated $CO_2$ alone: *Limacina antarctica* (789-1000 µatm,
24 h: Seibel et al., 2012) and *Diacria quadridentata* (1000 µatm, 6-18 h: Maas et al., 2012a). The
metabolic rates of all other species yet studied, including *Hyalocylis striata, Clio pyramidata,*
*Diacavolinia longirostris, Creseis virgula* (6-18 h: Maas et al., 2012a), and *Limacina helicina*
(24 h: Comeau et al., 2010a), were not significantly affected by short term exposure to high $CO_2$,
although the latter species showed an increase in metabolic rate when high $CO_2$ was combined
with high temperatures. Our results, which increase the geographic coverage for *L. helicina* and
*C. pyramidata* and provide the first data about the species *C. pacifica, C. atlantica, L.retroversa,*
*D. trispinosa, C.inflexa*  and *S. subula,* corroborate these earlier findings.
One interpretation of these results is that physiological responses may have occurred, but
involved the reallocation of resources to different tissues or metabolic pathways; this
redistribution could serve to maintain the thecosome total energy budget, and subsequently
would not significantly change the metabolic rate of the individuals. A transcriptomic study done
with individuals of *Clio pyramidata* as a companion project to the present work in fact suggested
that expression of some genes was influenced by $CO_2$ exposure even though metabolic rate is not
(Maas et al., 2015), perhaps suggesting some re-allocation among energetic demands. If this is
the case it indicates that, to some degree, the short-term exposure to high $CO_2$ concentration is



within the physiological tolerance of the tested species. Alternative hypotheses are that the
duration of exposure was too short or the severity of the $CO_2$ treatment too minimal to elicit a
measurable response. It is possible, for example, that some processes, like biomineralization,
may be influenced by high $CO_2$, but only after a longer exposure duration. Finally, it may be that
changes in respiration rate were subtle, requiring a much greater sample size to identify in light
of biological variability, but exploration of this hypothesis would require a dedicated experiment
to collect more individuals and likely a smaller number of species.

This possible tolerance to short term $CO_2$ exposure may be due to the fact that within

their distribution or diel migrational range there are conditions, or perhaps seasons, where the
natural hydrography causes many species of thecosome to experience conditions of high
$CO_2$/low pH, and the species are therefore adapted to this range of exposure. The Arctic species
*L. helicina* and subarctic species *L. retroversa*, for instance, are thought to inhabit waters which
have been shown to reach a concentration of $> 950$ µatm $CO_2$ and to be undersaturated with
respect to aragonite during the winter season in Kongsfjord, Svalbard (Lischka and Riebesell,
2012). These conditions are pervasive throughout the upper water column, meaning that *L.*
*helicina* and *L. retroversa*, which are not strong diel migrators, would experience seasonal under-
saturation in these polar regions. The more temperate and tropical open ocean thecosomes,
including *C. pyramidata, C. inflexa* and *S. subula* are all currently believed to be circumglobal
and most, to varying degrees, diel migratory (Table 1; Bé and Gilmer, 1977; van der Spoel,
1967). Populations are therefore likely to encounter high $CO_2$ in sub-surface waters in regions
associated with OMZs, including much of the North Pacific and off the coast of Northern Africa.
The ability to cope with high $CO_2$ for short durations may have been selected for over time as a
natural consequence of the types of unavoidable environmental variability experienced by these
planktonic populations.
**4.3 Low $O_2$ and Combined Effects**
In the Pacific Ocean, none of the species for which we had enough individuals to perform the
low $O_2$ study (*L. helicina, C. pyramidata,* and *C. inflexa*) had a significant change in metabolic
rate under low (10%) $O_2$, even when combined with enhanced $CO_2$. These results indicate that
the $O_2$ levels were above the concentration below which these species can no longer sustain their
routine metabolic activity (Pcrit; Hochachka and Somero, 2002) and that any changes in
physiology associated with the treatments required no increased energetic expenditure or



metabolic reduction. As subsurface waters throughout the cruise were frequently below 10% $O_2$
($< \sim$130 µmol kg$^{-1}$), this indicates that these species may be naturally adapted to coping with low
$O_2$ conditions.

In the Atlantic, examination of the effects of low $O_2$ is confound by an unfortunate and

accidentally low level of $CO_2$ ($\sim$130 µatm) in the LC/HO treatment (Table 3). Tests of the effect
of high $CO_2$ (HC/HO) and the interactive (HC/LO) treatments nonetheless remain valid, and for
*L. retroversa*, exposure to HC/LO caused a large and significant reduction in metabolic rate.
Suppression in metabolic rate is a common tactic for surviving unfavorable conditions (Guppy
and Withers, 1999; Seibel, 2011). Although metabolic depression is generally survivable in the
short term, over longer time scales there are often implications for growth, reproduction and
survival (reviewed in: Pörtner, 2010; Seibel, 2011). In the Atlantic, our measured in situ $O_2$
levels were never below 15% ($\sim$200 µmol kg$^{-1}$).  In contrast with the other species studied, which
in at least some portions of their geographic range are occasionally found in association with
subsurface low $O_2$ combined with hypercapnia, *L. retroversa* lives exclusively in the sub-polar
North Atlantic Ocean and the Southern Circumpolar Current. As such this is the only species in
this study in which no population is likely to experience conditions of low $O_2$ and high $CO_2$
together naturally anywhere in its distribution. Its inability to maintain metabolic rate during this
interactive exposure may be a short-term metabolic response to environmental conditions that are
unsustainable over longer time periods. As a consequence of the very low $CO_2$ in the LC/LO
treatment, it is impossible to determine whether the metabolic suppression for *L. retroversa* in
the HC/LO was in response to reduced $O_2$ availability alone or to the interactive effect of low $O_2$
with high $CO_2$. In the LC/LO treatment any change in respiration due to low $O_2$ could have been
masked by a change in the energy budget as a response to the low (equivalent to pre-industrial
atmospheric conditions) levels of $CO_2$. The results suggest that further work in the Atlantic is
warranted to disentangle these stressors and to determine whether the observed change in
metabolic rate was solely a consequence of $O_2$ availability or truly a synergistic effect.

Interestingly, although the temperature coefficients were not species-specific and may

not, therefore, perfectly normalize the dataset, one trend revealed by their use was a significant
difference in the normalized metabolic rates between species such as *S. subula* and *C. inflexa*
from the Atlantic and Pacific Oceans. The comparatively lower metabolic rates from the Pacific
may be a real response to the lower availability of $O_2$ for aerobic metabolism. Having a slower



routine rate of $O_2$ consumption may be the result of a more efficient respiratory mechanism or an
adaptation for coping with occasional exposures to the relatively high $CO_2$ and low $O_2$ conditions
found in the northeast Pacific Ocean.

**5. Conclusions**
Thecosomes pteropods are thought to be some of the most sensitive of the oceanic zooplankton
species to acidification. The responses we documented in the face of short-term $CO_2$ exposure
and low $O_2$ reveal interesting patterns about basin scale differences in sensitivity, possibly
associated with adaptation to local environmental conditions. Importantly, our results indicate
that short-term exposure to high $CO_2$ does not have an effect on the respiration rate of multiple
species of temperate and sub-polar thecosome species from both the North Atlantic and Pacific
Oceans, irrespective of recent likely environmental exposure. The lack of effect of $CO_2$ as a
single-stressor on metabolic rate in adult organisms of various species has been seen in a number
of studies (reviewed in: Dupont et al., 2010; Kroeker et al., 2013), although there are many other
metrics that have been shown to be more consistently affected. As such, thecosomes may have
physiological coping mechanisms that allow them to maintain their energy budget for short
periods of time in the face of high $CO_2$ via the re-allocation of their energetic resources. Over
longer time periods, however, this could reduce their scope for growth and reproduction,
negatively impacting the fitness of the population as has been demonstrated with other marine
calcifiers (i.e.: Dupont et al., 2013; Melzner et al., 2013; Stumpp et al., 2011). Testing these
hypotheses remains difficult as thecosomes are hard to maintain in captivity and there are no
published studies of individuals kept fed and exposed to $CO_2$ in laboratory conditions for long
durations (reviewed in: Howes et al., 2014; Thabet et al., 2015). Keeping individuals well fed is
a critical factor since high food availability has been suggested to modulate the effect of high
$CO_2$ exposure in both thecosomes (Seibel et al., 2012) and in other calcifying species (Thomsen
et al., 2013). Comparative short-term studies of wild caught animals such as the present
experiments, therefore, currently give us the best insight into the sensitivity of these open-ocean
populations, and the ability to predict how they will respond to the expected changes in the ocean
environment.

These findings also draw attention to the consequences of the high degree of vertical

variability in the open ocean environment, with animals in the Pacific found migrating between



679 deep waters, undersaturated with respect to aragonite, and the surface (Lawson, unpublished

680 data; Chu et al., in review; Maas et al., 2012a). Recent studies in the California Current system

681 indicate that thecosome shells show signs of in situ dissolution when associated with water

682 masses that are undersaturated with respect to aragonite (Bednaršek et al., 2014b; Bednarsek and

683 Ohman, 2015). Although our short duration experiments do not directly address the effect of

684 longer-term exposure to high $CO_2$, it does remind us that as open ocean environments respond to

685 anthropogenic change there may be vertical refugia from OA stress as well as regions where

686 animals may already experience high $CO_2$. As surface waters acidify, the ability to endure short-

687 duration exposure and to migrate in both the Atlantic and Pacific populations may provide

688 mechanisms for mitigating detrimental effects of acidification exposure. The potential

689 compression of vertical habitat associated with the shoaling of the aragonite compensation depth,

690 however, may have implications for predator/prey interactions, carbon pumping and other

691 ecosystem functions (Bednarsek and Ohman, 2015; Seibel, 2011). Furthermore, it is clear that

692 thecosome shells are highly sensitive to dissolution (Comeau et al., 2012; Lischka and Riebesell,

693 2012; Manno et al., 2012) and there could be fitness and ecological consequences of dissolution

694 in regions with vertical variation in carbonate chemistry.

695  Finally, as concerns about increasing $CO_2$ drive further explorations of comparative

696 organismal physiology in the marine system, it is important to recognize that often the exposure

697 of animals to increased $CO_2$ will occur in concert with expanding regions of low $O_2$. This has

698 been explored in the coastal environment where the interaction of acidification with

699 eutrophication and associated low $O_2$ is comparatively well studied (Cai et al., 2011; Melzner et

700 al., 2013), and in theoretical frameworks (Gruber, 2011; Pörtner, 2010; Sokolova, 2013).

701 Experiments in the open ocean environment, however, are only beginning to be conducted and

702 their implications explored. This study suggests that to make accurate predictions about how

703 populations will respond to climate change and adequately understand the factors affecting

704 organismal response, further investigations of the interactive effects of low $O_2$ and hypercapnia

705 should consider natural environmental variability, population biogeography and phylogenetic

706 sensitivity.



**Data availability**


Cruise data for the project is available via BCO-DMO under the project "Horizontal and Vertical
Distribution of Thecosome Pteropods in Relation to Carbonate Chemistry in the Northwest
Atlantic and Northeast Pacific" (http://www.bco-dmo.org/project/2154). The raw data for the
respiration experiments are included in this deposition (DOI: 10.1575/1912/6421).

**Author contributions**
A. Maas and G. Lawson designed the experiments. All co-authors participated in oceanographic
cruises and collection of samples. A. Maas conducted all of the experiments and statistical
analyses. Z.A. Wang advised on the design of the carbonate chemistry analysis and provided the
measurements of both the hydrographic and experimental conditions. A. Maas prepared the
manuscript with contributions from all co-authors.

**Acknowledgements**
We would like to acknowledge the hard work and dedication of the Captains and crews of both
the R/V *Oceanus* and R/V *New Horizon*, and to thank all the scientists, students and volunteers
who participated in the research expeditions. We are grateful to Brad Seibel, Scott Gallager, and
Dan McCorkle for lending us equipment. We would also like to thank Leocadio Blanco Bercial,
Peter Wiebe, Nancy Copley, Sophie Chu and Katherine Hoering for their support, insight and
input into methodologies, analysis and interpretation. Andy Solow kindly assisted with the
statistical model and interpretation. This work was funded by the National Science Foundation's
Ocean Acidification Program (grant OCE-1041068), National Institute of Standards and
Technology (NIST-60NANB10D024) and the WHOI postdoctoral scholarship program.





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

| Species | (optimal) temp (°C) | (optimal), depth (m) | migrator? |
|---|---|---|---|
| *Cuvierina atlantica* | 18 to 26 | 100-250 | possible |
| *Cuvierina pacifica* | Only recently established as a separate species, the habits are assumed to be similar to the Atlantic congener. | | |
| *Cavolinia inflexa* | 16 to 28 | 0-250 | no |
| *Clio pyramidata* | 7 to 27 | (0-500), <1500 | yes |
| *Limacina helicina* | (-2 to 10) | (50-100), <300 | possible |
| *Limacina retroversa* | (7 to 12) | (20-30), < 150 | possible |
| *Styliola subula* | (18 to 22) | 50-300 | yes |
| *Diacria trispinosa* | 9 to 28 | 30-200 | no |




Table 2: The hydrography and location for each station where animals for experiments were
collected. At stations along the main transect the depth (m) at which $O_2$ decreased below 130
$\mu$mol $O_2$ kg$^{-1}$ (~10%) and the average temperature from 25-100 m (°C) were derived from CTD
casts. At a few stations (denoted via $^a$) in the Atlantic there was warm water at the surface and
cold water below. The only species in this region, *Limacina retroversa*, has an optimum
temperature between 7-12 °C (Bigelow, 1924) and was generally found above 50 m (Lawson,
unpublished data). At these sites the average temperature is reported first for between 25-100 m
and then also for 25-50 m to reflect the conditions likely experienced by the pteropods. $pCO_2$ and
$\Omega_{Ar}$ were calculated from measured pH and DIC bottle samples. We interpolated linearly the
depths (m) at which the pH decreased below 7.7, $pCO_2$ reached 800 $\mu$atm, and aragonite
saturation ($\Omega_{Ar}$) reached 1 from the discrete measurements at adjacent depths. At stations
conducted while in transit to the main study transects (denoted by prefix T) the average
temperature from 25-100 m (°C) was documented from XBT casts. At these transit stations no
$O_2$ or carbonate chemistry data were available (noted with a dash). The species caught at each
station and used in this study are demarcated with a star (*).





| Year | Station | Latitude (°N) | Longitude (°W) | average temp 25-100 m | depth of 130 μmol O$_2$ kg$^{-1}$ | depth of pH 7.7 | depth of 800 μatm | depth of Ω$_{Ar}$ = 1 | C. atlantica | C. pacifica | C. inflexa | C. pyramidata | L. helicina | L. retroversa | S. subula | D. trispinosa |
|---|---|---|---|---|---|---|---|---|---|---|---|---|---|---|---|---|
| 2011 Atlantic | 32 | 49.1 | -44.3 | 5.3, 9.0 | NA | 74.1 | NA | NA | | | | | | * | | * |
| | 31 | 50.0 | -42.0 | 14 | NA | 385.4 | NA | NA | * | | | * | | | | * |
| | 30 | 49.6 | -41.9 | 14.1 | NA | 452.8 | NA | NA | * | | | * | | | | |
| | 26 | 47.5 | -42.0 | 13.3 | NA | 644.9 | NA | NA | * | | | * | | | | * |
| | 24 | 46.5 | -42.0 | 14.5 | NA | 453.9 | NA | NA | | | | | | | | |
| | 21 | 44.9 | -42.0 | 16.5 | NA | 501.1 | NA | NA | | | | | | | | |
| | 19 | 44.0 | -44.9 | 4.9, 11.2 | NA | 181.0 | NA | NA | * | | * | | | * | * | |
| | 17 | 43.0 | -47.8 | 1.8, 8.1 | NA | 143.1 | NA | NA | * | | * | | | * | * | |
| | 13 | 40.9 | -52.0 | 20.7 | NA | 756.7 | NA | NA | * | | * | | | | * | |
| | 10 | 47.5 | -52.0 | 19.4 | NA | 466.9 | NA | NA | | | | | | | | |
| | 8 | 38.5 | -52.0 | 22.8 | NA | 805.7 | NA | NA | | | | | | | | |
| | 3 | 36.0 | -52.0 | 21.4 | NA | 937.7 | NA | NA | * | | | | | | | |
| 2012 Pacific | T2 | 45.6 | -128.5 | - | - | - | - | - | | | | * | * | | | |
| | T3 | 46.6 | -133.5 | - | - | - | - | - | | | | | * | | | |
| | T4 | 47.7 | -138.5 | 6.4 | - | - | - | - | | | | | | | | |
| | T5 | 45.7 | -129.8 | 10.0 | - | - | - | - | | | | | * | | | |
| | T6 | 46.6 | -134.9 | 9.5 | - | - | - | - | | | | | * | | | |
| | T7 | 47.6 | -140.2 | 8.6 | - | - | - | - | | | | | | | | |
| | 3 | 49.0 | -148.2 | 6.2 | 209 | 128.9 | 193.7 | 168.5 | | | | * | * | | | |
| | 6 | 47.5 | -145.6 | 7.1 | 235 | 108.3 | 199.2 | 159.1 | | | | | | | | |
| | 7 | 47.0 | -144.6 | 7.8 | 256 | 131.0 | 214.0 | 185.1 | | | | * | | | | |
| | 15 | 43.1 | -138.1 | 10.9 | 363 | 199.5 | 368.2 | 334.8 | | * | | * | | | | |
| | 18 | 41.5 | -135.8 | 13.7 | 340 | 147.3 | 331.7 | 380.6 | | * | | * | | | | |
| | 21 | 39.9 | -135.0 | 12.7 | 348 | 162.0 | 332.2 | 302.8 | | * | | * | | | | |
| | 24 | 38.6 | -135.0 | 14.7 | 402 | 222.8 | 411.8 | 372.7 | | * | * | * | | | | |
| | 30 | 35.6 | -135.0 | 16.2 | 349 | 200.7 | 437.8 | 425.1 | | | * | * | | | * | |
| | 32 | 34.4 | -135.1 | 16.5 | 348 | 202.9 | 439.2 | 432.0 | | * | * | * | | | | |
| | 34 | 33.6 | -135.0 | 17.4 | 368 | 233.3 | 370.1 | 352.4 | | * | * | * | | | | |
| | T9 | 33.7 | -133.6 | 17.0 | - | - | - | - | | | | * | | | | |
| | T10 | 33.8 | -133.2 | 15.9 | - | - | - | - | | | | * | | | | |





Table 3: Carbonate chemistry during manipulation experiments. The manipulation experiments were conducted at multiple temperatures (T.) and salinities (S.) based on the conditions the organisms were caught in. As described in more detail in the text, DIC measurements were made of water drawn from the control chambers while TA was measured for batches of experimental water (denoted as xpt. TA). In situ TA (i.s. TA), based on nearby CTD bottle sampling at the surface, is also shown. At test stations, where bottle samples of in situ TA were unavailable, underway pCO$_2$ values and the LC/HO DIC were used to calculate in situ TA (denoted with *). In some instances, measurements of experimental TA differed by >20 μmol kg$^{-1}$ from nearby in situ measurements of surface TA. This difference greatly exceeds expected variability based on measurement uncertainty and spatial (geographic and vertical) offsets in the locations of experimental water collection relative to the nearest CTD cast; in these circumstances, the experimental TA was likely erroneous due to sampling errors (e.g., contamination). For completeness, and to aid in identification of erroneous experimental TA values, calculations of carbonate chemistry parameters, including aragonite saturation state (Ω$_{Ar}$) and pCO$_2$ were made based on DIC and both experimental TA and in situ TA. In further data analysis and interpretation, calculations based on experimental TA are given preference except those few instances where experimental TA differed from in situ by >20 μmol kg$^{-1}$ (bold denotes preferred calculations). Calculated saturation state and pCO$_2$ are reported as the average and standard deviation per batch of water. Note that the LC/LO gas tank in 2011 (in italics) appears to have been improperly mixed by the manufacturer as calculations suggested it contained a much lower CO$_2$ level than the intended 380 μatm; it should consequently be considered an entirely separate treatment from the 2011 LC/HO (were CO$_2$ levels were based on bubbling with an ambient air line).




| | Treatment | T. °C | S. | i.s. TA (µmol kg⁻¹) | xpt. TA (µmol kg⁻¹) | DIC (µmol kg⁻¹) | i.s. ΩAr | i.s. pCO₂ (µatm) | xpt. ΩAr | xpt. pCO₂ (µatm) |
|---|---|---|---|---|---|---|---|---|---|---|
| 2011 Atlantic | 380 µatm CO₂ / 21% O₂ | 10 | 33 | 2300.3 | **2307.3** | 2094.4 | 2.3 ± 0.2 | 336.2 ± 37.7 | **2.4 ± 0.2** | **324.8 ± 35.8** |
| | | 15 | 33 | 2300.3 | **2307.3** | 2066.5 | 2.6 ± 0.7 | 404.5 ± 172.7 | **2.7 ± 0.7** | **390.8 ± 164.5** |
| | | 15 | 35 | **2296.4** | 2354.5 | 2066.4 | **2.5 ± 0.1** | **382.3 ± 20.4** | 3.1 ± 0.1 | 297.7 ± 14.3 |
| | | 20 | 34 | 2353.4* | **2345.8** | 2028.6 | 3.6 ± 0.2 | 302.8 ± 31.6 | **3.5 ± 0.2** | **311.6 ± 32.9** |
| | | 20 | 34 | 2366.0 | **2367.2** | 2077.5 | 3.3 ± 0.1 | 363.1 ± 23.2 | **3.3 ± 0.1** | **361.4 ± 23.1** |
| | *380 µatm CO₂ / 10% O₂* | 10 | 33 | 2300.3 | **2307.3** | 1919.7 | 4.0 | 139.0 | 4.1 | **135.5** |
| | | 15 | 33 | 2300.3 | **2307.3** | 1774.8 | 5.5 ± 0.6 | 101.2 ± 23.9 | **5.6 ± 0.6** | **99.0 ± 23.3** |
| | | 15 | 35 | **2296.4** | 2354.5 | 1852.7 | **4.6** | **139.2** | 5.3 | 116.1 |
| | 800 µatm CO₂ / 21% O₂ | 10 | 33 | 2300.3 | **2307.3** | 2219.7 | 1.2 ± 0.2 | 779.9 ± 114.0 | **1.2 ± 0.2** | **742.4 ± 106.8** |
| | | 15 | 33 | 2300.3 | **2307.3** | 2208.0 | 1.3 | 908.7 | 1.4 | **867.8** |
| | | 15 | 35 | **2296.4** | 2354.5 | 2139.5 | **1.9** | **585.2** | 2.4 | 434.4 |
| | | 20 | 34 | 2353.4* | **2345.8** | 2176.9 | 2.1 ± 0.1 | 651.8 ± 23.4 | **2.1 ± 0.1** | **678.2 ± 24.8** |
| | | 20 | 34 | 2366.0 | **2367.2** | 2212.7 | 1.9 ± 0.4 | 786.0 ± 196.0 | **1.9 ± 0.4** | **780.9 ± 194.2** |
| | 800 µatm CO₂ / 10% O₂ | 15 | 33 | 2300.3 | **2307.3** | 2186.2 | 1.5 ± 0.2 | 788.7 ± 157.6 | **1.5 ± 0.2** | **754.9 ± 148.3** |
| | | 15 | 35 | **2296.4** | 2354.5 | 2179.6 | **1.5 ± 0.3** | **782.9 ± 164.6** | 2.0 ± 0.3 | 558.2 ± 103.9 |
| 2012 Pacific | 380 µatm CO₂ / 21% O₂ | 10 | 32.1 | 2151.9* | **2142.8** | 1934.8 | 2.2 ± 0.1 | 285.2 ± 21.4 | **2.3 ± 0.1** | **283.0 ± 21.2** |
| | | 10 | 33.5 | 2208.0 | **2222.7** | 2001.9 | 2.4 ± 0.6 | 302.2 ± 100.9 | **2.4 ± 0.6** | **303.3 ± 101.4** |
| | | 15 | 32.5 | **2182.6*** | 2095.7 | 1983.4 | **2.2 ± 0.0** | **388.1 ± 5.5** | 1.4 ± 0.0 | 646.7 ± 11.5 |
| | | 15 | 33.5 | 2208.0 | **2222.7** | 2020.8 | 2.3 ± 0.2 | 407.7 ± 52.1 | **2.3 ± 0.2** | **409.1 ± 52.4** |
| | 380 µatm CO₂ / 10% O₂ | 10 | 32.5 | **2182.6*** | 2095.7 | 1973.9 | **2.3 ± 0.1** | **295.5 ± 20.0** | 1.4 ± 0.1 | 489.2 ± 41.2 |
| | | 15 | 33.5 | 2208.0 | **2222.7** | 2017.5 | 2.3 | 3956.0 | 2.3 | **397.4** |
| | 800 µatm CO₂ / 21% O₂ | 10 | 32.1 | 2151.9* | **2142.8** | 2026.3 | 1.4 ± 0.1 | 525.0 ± 35.0 | **1.4 ± 0.1** | **519.7 ± 34.5** |
| | | 10 | 33.5 | 2208.0 | **2222.7** | 2120.6 | 1.3 | 628.2 | 1.3 | **631.2** |
| | | 15 | 32.5 | **2182.6*** | 2095.7 | 2031.7 | **1.8 ± 0.1** | **527.6 ± 50.9** | 1.0 ± 0.1 | 952.4 ± 115.1 |
| | | 15 | 33.5 | 2208.0 | **2222.7** | 2112.2 | 1.4 ± 0.2 | 736.0 ± 96.0 | **1.4 ± 0.2** | **739.4 ± 96.6** |
| | 800 µatm CO₂ / 10% O₂ | 10 | 32.5 | **2182.6*** | 2095.7 | 2066.5 | **1.4 ± 0.1** | **545.5 ± 65.1** | 0.8 ± 0.1 | 1056.0 ± 151.6 |
| | | 15 | 33.5 | 2208.0 | **2222.7** | 2118.3 | 1.4 | 762.4 | 1.4 | **766.0** |





Table 4: The average wet mass (mass; g) and mass-specific oxygen consumption rate ($MO_2$; $\mu$mol $O_2$ $g^{-1}$ $h^{-1}$) ± the standard errror (SE) for each treatment (Treat.) and species. The number of individuals (N) per treatment are reported and the species are arranged by temperature (Temp; °C) as well as the year and basin of collection.

| Year | Temp. | Species | Treat. | N | mass | ±SE | $MO_2$ | ±SE |
|---|---|---|---|---|---|---|---|---|
| 2011 | 10 | *Limacina retroversa* | LC/HO | 12 | .00281 | 0.00037 | 10.33 | 1.17 |
| Atlantic | | | HC/HO | 13 | .00284 | 0.00031 | 10.10 | 0.56 |
| | | | LC/LO | 9 | .00274 | 0.00026 | 8.12 | 0.66 |
| | | | HC/LO | 9 | .00377 | 0.00053 | 4.21 | 0.55 |
| | 15 | *Clio pyramidata* | LC/HO | 10 | .01944 | 0.00408 | 7.81 | 0.71 |
| | | | HC/HO | 8 | .01410 | 0.00435 | 8.55 | 1.48 |
| | | | LC/LO | 9 | .02363 | 0.00867 | 6.63 | 1.21 |
| | | | HC/LO | 8 | .03945 | 0.00467 | 6.99 | 0.45 |
| | | *Cuvierina atlantica* | LC/HO | 8 | .04493 | 0.00264 | 5.05 | 0.63 |
| | | | LC/LO | 10 | .04636 | 0.00252 | 3.25 | 0.28 |
| | | | HC/LO | 10 | .05040 | 0.00219 | 4.29 | 0.37 |
| | | *Diacria trispinosa* | LC/HO | 8 | .03718 | 0.00316 | 4.44 | 0.56 |
| | | | HC/HO | 10 | .03589 | 0.0027 | 4.09 | 0.51 |
| | 20 | *Cuvierina atlantica* | LC/HO | 9 | .01876 | 0.00396 | 4.31 | 0.85 |
| | | | HC/HO | 9 | .01683 | 0.00284 | 4.53 | 1.13 |
| | | *Cavolinia inflexa* | LC/HO | 8 | .00626 | 0.00104 | 14.30 | 1.48 |
| | | | HC/HO | 4 | .00508 | 0.00049 | 13.81 | 1.39 |
| | | *Styliola subula* | LC/HO | 10 | .00400 | 0.00038 | 13.96 | 1.80 |
| | | | HC/HO | 8 | .00289 | 0.00035 | 15.95 | 0.87 |
| 2012 | 10 | *Limacina helicina* | LC/HO | 7 | .00140 | 0.00026 | 5.26 | 1.17 |
| Pacific | | | HC/HO | 8 | .00149 | 0.00021 | 5.51 | 0.69 |
| | | | LC/LO | 6 | .00300 | 0.00058 | 4.91 | 0.69 |
| | | | HC/LO | 10 | .00296 | 0.00038 | 7.18 | 1.45 |
| | | *Clio pyramidata* | LC/HO | 9 | .02646 | 0.00258 | 5.43 | 0.45 |
| | | | HC/HO | 8 | .02355 | 0.00369 | 4.39 | 0.60 |
| | | | LC/LO | 14 | .01459 | 0.00185 | 5.58 | 0.81 |
| | | | HC/LO | 12 | .01250 | 0.00245 | 5.72 | 1.14 |
| | 15 | *Cuvierina pacifica* | LC/HO | 4 | .01829 | 0.00563 | 3.41 | 0.56 |
| | | | HC/HO | 7 | .02130 | 0.00636 | 3.53 | 0.57 |
| | | *Cavolinia inflexa* | LC/HO | 5 | .01330 | 0.00062 | 3.53 | 0.44 |
| | | | HC/HO | 8 | .01556 | 0.00149 | 3.34 | 0.41 |
| | | | LC/LO | 4 | .01405 | 0.00185 | 2.41 | 0.33 |
| | | | HC/LO | 2 | .01855 | | 3.98 | |
| | | *Styliola subula* | LC/HO | 6 | .00360 | 0.00044 | 5.30 | 1.20 |
| | | | HC/HO | 4 | .00220 | 0.00029 | 7.73 | 2.14 |
| | | *Clio pyramidata* | LC/HO | 4 | .03020 | 0.0037 | 3.82 | 0.66 |
| | | | HC/HO | 5 | .02904 | 0.00329 | 3.21 | 0.27 |



Table 5: Statistical results of the univariate general linear models (GLM) for each species were analyzed separately by year and are listed by the temperature of the experiment (Temp.; °C). For species studied at multiple temperatures (denoted by *), the metabolic rates were adjusted to 15°C using a $Q_{10} = 2$ to allow for direct comparison. The effect of the independent factors of $CO_2$ level ($CO_2$), $O_2$ level ($O_2$), their interactive effect (Int.) and the covariate of mass were analyzed in regards to the metabolic rate and reported as *p*-values for the Pacific (mean mass specific metabolic rate values found in Table 4). For the Atlantic, each treatment was tested as independent (Treat.) due to the accidentally low $CO_2$ condition in the LC/LO gas mixture.

| Year | Temp. | Species | $CO_2$ | $O_2$ | Int. | Treat. | Mass |
|------|-------|---------|--------|-------|------|--------|------|
| | | | | | | Effect on metabolic rate | |
| 2011 | 10 | *Limacina retroversa* | | | | **<0.001** | <0.001 |
| Atlantic | 15 | *Clio pyramidata* | | | | 0.295 | <0.001 |
| | | *Cuvierina atlantica** | | | | 0.174 | <0.001 |
| | | *Diacria trispinosa* | .731 | | | | <0.001 |
| | | *Cavolinia inflexa* | .677 | | | | .008 |
| | | *Styliola subula* | .791 | | | | .040 |
| 2012 | 10 | *Limacina helicina* | .464 | .323 | .914 | | .007 |
| Pacific | 15 | *Clio pyramidata** | .255 | .156 | .726 | | .018 |
| | | *Cuvierina pacifica* | .709 | | | | <0.001 |
| | | *Cavolinia inflexa* | .309 | .717 | .219 | | .113 |
| | | *Styliola subula* | .763 | | | | .668 |





**Figure legends**

**Figure 1: Cruise tracks and animal sampling.** Thecosomes were collected during the night at stations along the main survey transect (solid line) and at stations during transit (dashed line) during cruises to the northwest Atlantic in 2011 and northeast Pacific in 2012. The shapes correspond to the species caught at each station and used in this study. Blue (10 °C), grey (15 °C) and red (20 °C) boxes around the station numbers (#) correspond to the temperature that was representative of 25-100 m at each station (Table 2) and used in the experiments with animals from that station.

**Figure 2: Hydrography of sampling regions.** Hydrographic profiles of stations representative of the specific water mass types from the northern (P-T5, P-6, A-26), middle (P-18, A-19) and southern (P-32, A-8) portions of the Pacific (P) and Atlantic (A) study transects (station locations: Fig. 1). At station P-T5, the temperature profile (grey) was from an XBT cast because no CTDs were conducted during transits. For all stations along the main transects, left-hand plots show temperature (grey), salinity (black) and oxygen (black dotted) measured via sensors on the CTD and binned to 1 m depth intervals. Middle plots show TA (black) and DIC (grey) from from discrete bottle samples (dots show depths of bottle samples). Right-hand plots show $pCO_2$ (black) and aragonite saturation state ($\Omega$; grey) calculated based on TA and DIC measurements.

**Figure 3: Thecosome respirometry.** Mean metabolic rate and standard error ($\mu$mol $O_2$ g$^{-1}$ h$^{-1}$) of thecosomes exposed to low (i.e., ambient) $CO_2$ and normal levels of $O_2$ (light blue; LC/HO), high $CO_2$ and normal $O_2$ levels (dark blue; HC/HO), low $CO_2$ and low $O_2$ (light red; LC/LO), or high $CO_2$ and low $O_2$ (dark red; HC/LO). The species and temperature of the experiment are reported below the x-axis. Significance is reported based on a basin, species, and temperature specific GLM which tested for the effect of treatment on $O_2$ consumption with a Bonferroni post-hoc analysis. In the Atlantic analysis each treatment was tested independently, while in the Pacific $CO_2$ and $O_2$ were treated as factors. For each species and temperature, treatments are reported as non-significant (N.S.) or, in the case of significance, by letters that indicate which treatments are statistically similar (same letter) or different (different letter) at a p-value < 0.05.



Note that for *C. atlantica* the metabolic rates of individuals respired at 20° C were converted to 15°C using a temperature coefficient of 2 (see methods) for this GLM analysis.

**Figure 4:** Log transformed metabolic rates ($\mu$mol $O_2$ $h^{-1}$) for *L. retroversa* at 10 °C, not normalized to mass, plotted against the log transformed wet mass (mg) of individuals exposed to low $CO_2$ and normal levels of $O_2$ (black circles; LC/HO), high $CO_2$ and normal $O_2$ levels (dark grey diamonds; HC/HO), low $CO_2$ and low $O_2$ (white circles; LC/LO), or high $CO_2$ and low $O_2$ (light grey diamonds; HC/LO).





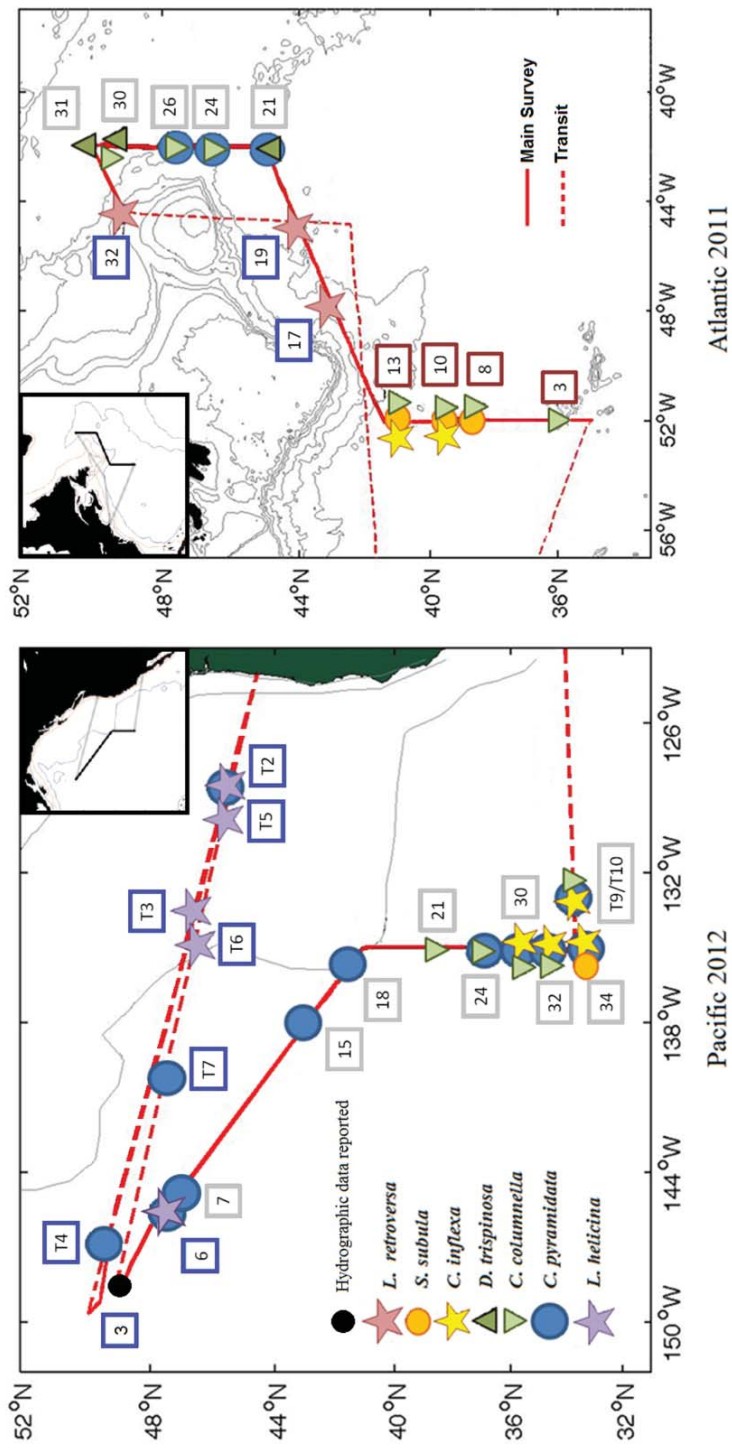





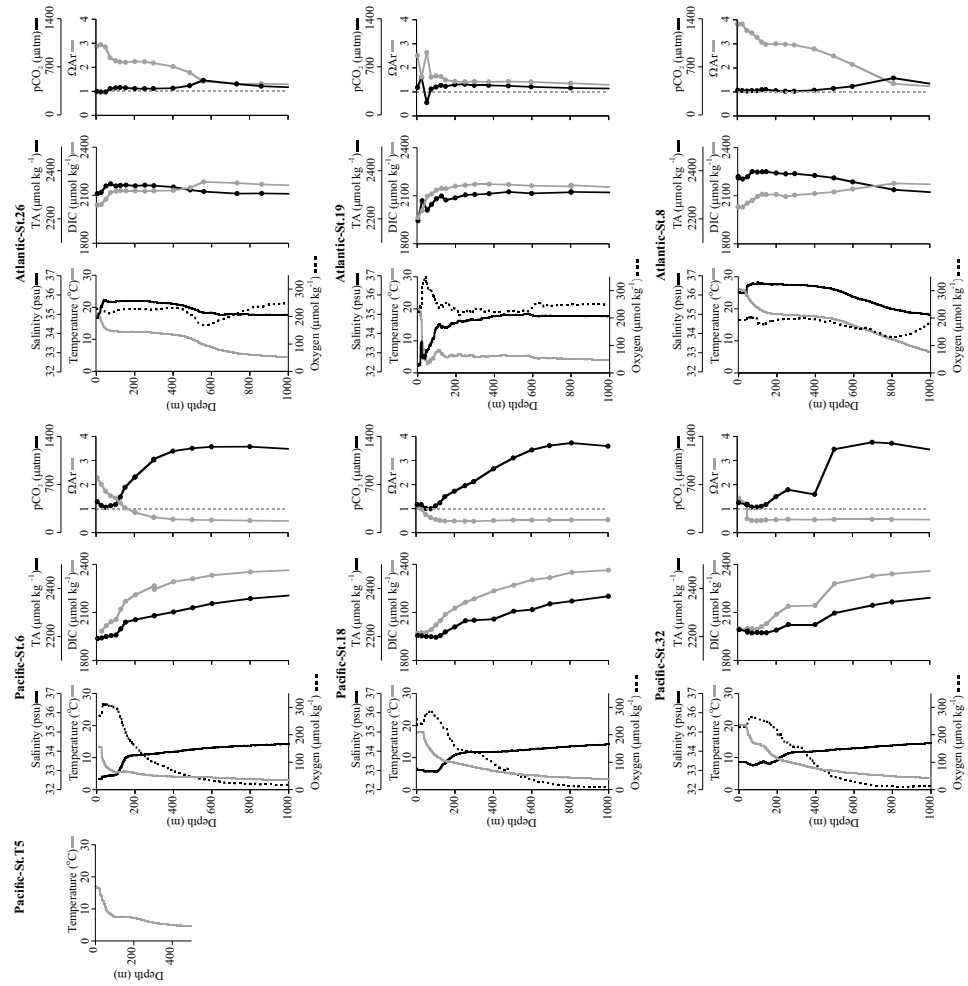









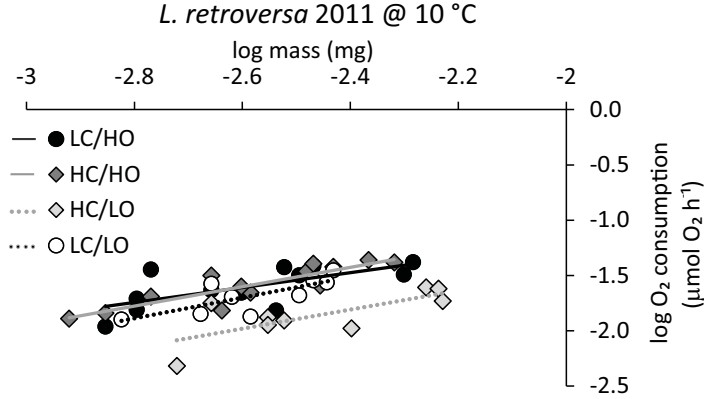