# Peer review of "The metabolic response of thecosome pteropods from the North Atlantic and North Pacific Oceans to high $CO_2$ and low $O_2$"

_Biogeosciences, 2016_

## Referee Comment (RC1) · Anonymous Referee #1 · 22 Jun 2016

Manuscript under review for Biogeoscience bg-2016-230

"The metabolic response of thecosome pteropods from the North Atlantic and North Pacific Oceans to high CO2 and low O2" by Maas AE, Lawson GL, Wang ZA

General comments

This manuscript deals with the question whether thecosome pteropods that naturally experience high CO2/low O2 conditions in their North Pacific habitat show different physiological response as compared to congeners from the North Atlantic were such hypercapnic and low O2 conditions do not prevail in depth ranges of pteropods' occurrence. In a set of experiments with different treatment combinations and short-

term exposure of pteropods, the authors performed oxygen consumption measurements with eight different pteropod species from both ocean regions and tested their response to high CO2 and low O2 conditions. The only species that showed a change in metabolic response was Limacina retroversa from the North Atlantic. This species reduced metabolic rates in response to the combined treatment (high CO2/low O2) but not to high CO2 alone. The authors conclude that pteropods have mechanisms to cope with short-term CO2 exposure and their considerations should be taken into account in projections for pteropods under future ocean conditions.

This is important work that contributes to our understanding of different pteropod species physiology and their possible response to ongoing ocean change, i.e. ocean acidification and extension of oxygen minimum zones due to further ocean warming. A clear strength is the use of a variety of pteropod species naturally experiencing high CO2 / low O2 histories to give better insight into how these species are adapted to these conditions. Not only from this perspective but also owing to the fact that there are still many open questions especially with respects pteropods physiology, this is important work that I will like to see published. I have some minor comments and criticism, however, that should be dealt with prior publication. Mostly this applies to clarification with respect to replication of the experiments, mentioning of data in the text and respective tables, as well as some more information on the actual life stage of pteropods worked with.

Specific comments

Material and Methods

- Section 2.4: How many organisms were incubated per respiration chamber, how many replicates were possible to set up? Is N in table 4 the number of replicates? Please clarify. Please give also size ranges of different species incubated. Of what stage were the pteropods (all adults?)? If different stages, how could that affect results and conclusions drawn?

- L319: Size ranges of all species would be really helpful to see!

- Section 2.6: Assumptions proved in case of significant results found for L. retroversa?

Results

- Section 3.1: Please include information on the size ranges of the different pteropod species. Were all individuals of the same developmental stage?

- Section 3.2: This section needs some clarification with respects mentioning of geographical positions, temperatures... in the text and the respective table.

- L409/410: Can not find geographical position in table 2?

- L411: 250 or 209 m?

- L412: 110 or 130 m?

- L415: geographical position not found in Table 2?

- L417: 10–17°C?

- L421–425: How can I see that Clio pyramidata experienced these conditions (in Table1)?

- L427: 200 $\mu$mol kg-1 corresponds to what % air saturation O2?

- L428/429: Would it be possible to indicate the dominant hydrographic regimes in Fig. 1, would be helpful in connection with the sampling location of different pteropod species.

- L432/433: 400 or 385 m? geopgraphical position can not be found in Table 1.

- L437: below 5°C? I calculated 5.2°C?

- Could salinity be included in Table 2?

- L478/480: This sentence is unclear. According to Table 3, $\Omega$ar was never below 1?

And 1.2 is not under-saturated? Please clarify!

- L482: In situ values are meant here, right? Maybe indicate in the text, easier for the reader.

- Section 3.4: As indicated earlier, please clarify how many replicates were measured and how many individuals were incubated per chamber and experiment and species.

- L526: Fig 4 not 4A

Discussion:

The discussion is in general ok, but I miss some consideration relating to the specific life stage(s) these experiments were done on. Was it all the same life stages? If not, could that affect results and conclusions?

- L552/553: What stage of L. helicina was it? Could the high mortality also be associated with life cycle issues and less so with temperature, i.e. die off after reproduction?

- L617: How likely is it that O2 saturation below 10% resulted in a substantial difference compared to the results obtained at 10% O2 saturation? In other words, any idea where a critical threshold level could lie?

Figures:

In general, please next time indicate numbers of figures and tables directly on the page were figures and tables are shown. The way they are presented here led to a continuous turning around of printed pages and searching for a particular one.

Table 4

- Please include a size range for all different species

- Is N consistent with number of replicates?

---

## Referee Comment (RC2) · Anonymous Referee #2 · 5 Jul 2016

This study deals with an interesting question: does life history (geographical origin) drive the response of pteropod metabolism to ocean acidification and low oxygen? This is a good manuscript that provides some critically needed data on the physiological response of pteropods to OA. While the presented data set is not perfect (missing treatments at some sites, problems with pCO2 manipulations), this approach looking at the combined effects of high pCO2/low O2 on pteropods originating from two distinct ocean basins is novel and adapted to look at potential acclimation. I have reviewed an earlier version of the manuscript and I am pleased to note that the authors took in consideration the previous comments. I have listed few specific comments below.

Comments: Abstract: The first sentence is a bit confusing. It reads like the burning of

fossil fuels directly cause a decrease in O2.

L72: This sentence is misleading and tends to indicate that with OA all oceans will be undersaturated.

L79: Probably the change in saturation state is not the only driver (DIC/proton ratios, pH itself, ..). I would replace this by "modifications of the carbonate chemistry".

L101: It would be good to compare this value with what is found in the other oceans.

L146-148: You mention it in the discussion, but it could be interesting to indicate here that they are potentially different species.

L-226: Add "Surface" before "carbonate chemistry".

L340: The effect is probably minor, but pteropod calcification and excretion can change the TA.

L350: Could the difference in TA be due to the bubbling that caused evaporation?

L456: Increased not decreased?

The results section contains a large part of methods and discussion. It reads well but I wonder if for clarity the methods and discussion statements should be moved to the corresponding sections.

---

## Author Comment (AC1) · 26 Aug 2016

**bg-2016-230**

**The metabolic response of thecosome pteropods from the North Atlantic and North Pacific Oceans to high CO2 and low O2**

A. E. Maas, G. L. Lawson, and Z. A. Wang

**We would like to thank the referee for their appreciation of our approach and their helpful comments for improving the clarity of the text. We have responded to each suggestion on a topically grouped point by point basis with referee comments in plain text and author response in bold.**

- Section 2.4: How many organisms were incubated per respiration chamber

**Each chamber contained one individual pteropod. The text has been modified to read:**

> **"Post-gut-clearance, healthy organisms were put into separate glass syringe respiration chambers, one individual per chamber, with a known volume of 0.2 μm filtered seawater and 25 mg L-1 each of streptomycin and ampicillin."**

how many replicates were possible to set up? Is N in table 4 the number of replicates? Please clarify.

- Is N consistent with number of replicates?

**The number of replicates was different for each species, treatment and region. The N in table 4 (previously labeled as the number of individuals) is indeed the number of replicates. In the case of this experimental design individuals and replicates are synonymous. The word 'individuals' has been replaced with 'replicates' for clarity.**

The discussion is in general ok, but I miss some consideration relating to the specific life stage(s) these experiments were done on. Was it all the same life stages? If not, could that affect results and conclusions?

Of what stage were the pteropods (all adults?)? If different stages, how could that affect results and conclusions drawn?

Were all individuals of the same developmental stage?

**To address this point, text is now included in the methods, results, discussion and conclusion specifically stating that the work was done only with adult life stages.**

> **"Species were targeted specifically for their abundance and the likelihood of their presence in both ocean basins and only adult individuals were used."**

> **"Only relatively large adult specimens were used in respiration trials, in part to avoid any confounding effects of ontogeny and in part to ensure a measurable change in oxygen levels."**

**We agree with the reviewer that juveniles of all the species we studied could potentially be more sensitive to the tested conditions. There is experimental evidence from our group that the veliger stage of _L. retroversa_ is more susceptible to $CO_2$ (Thabet et al. 2015) and clearly as projections are made about population-level responses to changing environmental conditions the sensitivities of all life stages must be considered. The vertical distribution of pteropod juveniles has not been well documented (in part because good morphological keys to that stage do not exist), but it would be interesting to determine whether they are absent from depths where high $CO_2$ / low $O_2$ is present naturally. Interestingly many diel vertically migratory species exhibit different vertical distribution throughout ontogeny in relation to midwater regions of high $CO_2$ / low $O_2$ (i.e. Wishner et al. 2013; Maas et al. 2014). It seems logical that juvenile pteropods would avoid regions of high $CO_2$ during initial shell formation as this is a period of sensitivity as has been shown in some pteropods (Thabet et al. 2015) and in other species (Kurihara 2008; Kroeker et al. 2013; Waldbusser et al. 2015).**

Since the text only focuses on adults we have chosen not to discuss this thoroughly; we did, however, reiterate our focus on the adult life stage at the start of the discussion:

"This study reveals that short term exposure to low $O_2$ and high $CO_2$, similar to what would be experienced by individuals in the Pacific during diel vertical migration, does not influence the oxygen consumption of adult individuals of most of the thecosome pteropod species examined from either the Atlantic or Pacific."

And made mention the importance of the consideration of the various sensitivities of different life stages as:

"Furthermore, although adult individuals may show no change in metabolic rate, there is evidence that juvenile stages of many calcifying species are typically more sensitive to $CO_2$ exposure (i.e. Connell et al. 2013; Waldbusser et al. 2015) and emerging evidence supports the idea that eggs, veligers and juveniles of *L. retroversa* and *L. helicina* are more vulnerable to acidification than adults (Lischka et al. 2011; Thabet et al. 2015; Manno et al. 2016). Thus, although adults may be capable of surviving short term exposure, as acidity in surface waters increase there may be population level stress due to ontogenetic sensitivity."

Please give also size ranges of different species incubated.
- L319: Size ranges of all species would be really helpful to see!
- Section 3.1: Please include information on the size ranges of the different pteropod species.
Table 4: - Please include a size range for all different species
The size range (average mass and standard error) for the pteropods is already reported in Table 4, and it seems unnecessary to report the range (smallest to largest). If the reviewer instead means size in length, the information is not available for some of the samples which have been used for other purposes. Furthermore, the species documented here vary substantially in shape and have different growth patterns, making it impossible to report a standard size (length/diameter, width). It seems that if length is what the reviewer refers to, this is tied into the previous concern about life stage and we hope that the changes made to the text are sufficient to allay the reviewers' concerns.

- Section 2.6: Assumptions proved in case of significant results found for L. retroversa?
We assume that the reviewer refers to the assumptions of normality and heterogeneity of variance – please correct if otherwise. These tests were performed and L. retroversa met all assumptions. Text has been added on line 370 as follows:

"The datasets were tested for normality and homoscedasticity." Information about the statistical assumptions has been added to Table 5.

- Section 3.2: This section needs some clarification with respects mentioning of geographical positions, temperatures. . . in the text and the respective table.
- L409/410: Can not find geographical position in table 2?
- L415: geographical position not found in Table 2?
- L432/433: 400 or 385 m? geopgraphical position can not be found in Table 1.
Text has been added to clarify the geographical positions of each station in the text (latitude, longitude and station number are now specifically described). The range reported in the text describes collectively all of the stations found in the portion of the study region, and hence is a summary of numbers found in Table 2 rather than re-stating exact values from that Table. In addition to the above changes, text has been added to the legend in table 2 to help point out the geographical locations of the hydrographic regimes. Line 1024:

"Each basin was characterized by multiple hydrographic regimes (see text and Fig 2); transitions between regimes are denoted by dashed horizontal lines."

- L411: 250 or 209 m?

**The sentence was intended to indicate that at all stations in this hydrographic regime, the oxygen fell below 130 within the range of the organisms. To better clarify this point the text has been changed to (Line 420-421):**

> **"At these stations O2 fell below 10% (~130 µmol kg$^{-1}$) at depths less than ~250 m"**

- L412: 110 or 130 m?

**It was between 108.3-131 m. To clarify the sentence was changed to (Line 421-422):**

> **"At these stations in the northern part of the transect, pH fell below 7.7 at depths less than 130 m,…"**

- L417: 10–17_C?

**The change has been made.**

- L421–425: How can I see that Clio pyramidata experienced these conditions (in Table1)?

**Table 1 reports that Clio pyramidata is a known vertical migrator with a typical range of 0-500 m, and can be found as deep as 1500 m. Based on this information, we would expect that its range similarly extends 500 m in the Pacific, which would put it into conditions of 10% O2, 800 µatm pCO2 and aragonite undersaturation in the Pacific as per the values documented in Table 2. To clarify we have referenced Table 2 in the sentence (which as a note has been moved to the discussion as per the request of reviewer 2).**

- L427: 200 µmol kg-1 corresponds to what % air saturation O2?

The information has been added to Line 436-438 as:

> **"In contrast to the Pacific, along the entire Atlantic transect O2 concentration was above ~200 µmol kg$^{-1}$ (~15%) in the top 500 m, while pCO$_2$ never reached 800 µatm and aragonite undersaturation never occurred throughout the top 1000 m."**

- L428/429: Would it be possible to indicate the dominant hydrographic regimes in
Fig. 1, would be helpful in connection with the sampling location of different pteropod species.

**An attempt was made to modify the figure, but there was no way to show the regimes clearly on what is already a busy figure. We hope that reviewer agrees that with the new additions to table 2 and the text there is now enough information to make this clear.**

- L437: below 5_C? I calculated 5.2_C?

**The sentence was intended to indicate that at all stations in this hydrographic regime, temperatures within the 25-100 m all reached below 5 C. The sentence has been modified as:**

> **"Stations conducted in this water were typified by a temperature and salinity anomaly with temperatures falling below 5°C from 25-100 m and a salinity signature < 33, contrasting significantly with the surface salinities of the northern portion (~34) and southern portion (~36) of the Atlantic transect.**

- Could salinity be included in Table 2?

**Salinity has been added to Table 2 and the table caption has been modified to reflect the change.**

- L478/480: This sentence is unclear. According to Table 3, ar was never below 1? And 1.2 is not under-saturated? Please clarify!

The point of the sentence was that omega reached a minimum of 1.2 and hence *approached*, but didn't reach, under-saturation. To clarify the sentence has been changed as follows:

"The experimental conditions of the high $CO_2$ treatments reached their lowest value in the middle part of the transect ($\Omega_{Ar}$ = 1.2 at mid-latitudes; Table 3), where cold northern waters of low salinity were encountered. Experimental $\Omega_{Ar}$ had a range of 1.5-2.0 for the rest of the transect in the Atlantic."

- L482: In situ values are meant here, right? Maybe indicate in the text, easier for the reader.
We meant the experimental and have now indicated this in line 492-494 as:

"The values of experimental $\Omega_{Ar}$ were lower overall in the Pacific, although the high $CO_2$ treatments also never reached under-saturation ($\Omega_{Ar}$ 1.3-1.8)."

- Section 3.4: As indicated earlier, please clarify how many replicates were measured and how many individuals were incubated per chamber and experiment and species.
The word 'individuals' has been replaced with 'replicates' in table 4 for clarity. We do not think it is appropriate to re-state that the method was to place single individuals in a chamber in the results, and hope that the reviewer agrees that the clarification in the methods and table are sufficient to clarify the point.

- L526: Fig 4 not 4A
The correction has been made

- L552/553: What stage of L. helicina was it? Could the high mortality also be associated with life cycle issues and less so with temperature, i.e. die off after reproduction?
It is possible, but based on our previous experiences with *Limacina spp.* these organisms are capable of laying eggs throughout their later adult life and typically don't die due to spawning (Maas pers. obs.; Thabet et al. 2015). It is true that they do eventually reach their largest size class and then die. An analysis of the available size class data of all the individuals does not support the idea that the older or younger (larger or smaller respectively – pteropods grow continuously throughout their lives) were differentially susceptible. Live individuals ranged in mass from 0.5-10.4 mg, while dead individuals were 0.9-5 mg. We did not measure individuals from the high mortality events, however, so size may have played a role? In any case, a sentence about alternative hypotheses has been added (Line 569-571):

"Alternative hypotheses are that these were population reaching senescence, or that they were collected in a hydrographic regime with low food availability."

- L617: How likely is it that O2 saturation below 10% resulted in a substantial difference compared to the results obtained at 10% O2 saturation? In other words, any idea where a critical threshold level could lie?
The lower than 10% $O_2$ saturation that was documented in the wild was never explicitly tested in our experiments. Our lowest experiment only ever reached ~8% oxygen over the course of the respiration. Since these were end point measurements there is no way to determine the $P_{crit}$. Based on previous observations we would hypothesize that the Pacific populations have a lower $P_{crit}$ than the Atlantic. Importantly some of these species in the Eastern Tropical Pacific have been shown to survive and respire at 1% $O_2$ (Maas et al. 2012). An explicit study of differences in $P_{crit}$ between the ocean basins using the more modern optical spot sensors (such as in (Kiko et al. 2016; Maas et al. 2016), would be informative and productive.

In general, please next time indicate numbers of figures and tables directly on the page were figures and tables are shown. The way they are presented here led to a continuous turning around of printed pages and searching for a particular one.

**We apologize that the reviewer found the lack of labels on the figures and tables confusing and will be sure to take this into consideration during our next submission.**

---

## Author Comment (AC2) · 26 Aug 2016

bg-2016-230
**The metabolic response of thecosome pteropods from the North Atlantic and North Pacific Oceans to high CO2 and low O2**
A. E. Maas, G. L. Lawson, and Z. A. Wang

**We would like to thank the referee for their helpful feedback on this, and a previous version, of this manuscript. We have responded to each suggestion on a point by point basis with referee comments in plain text and author response in bold.**

Comments:
Abstract: The first sentence is a bit confusing. It reads like the burning of fossil fuels directly cause a decrease in O2.
**The sentence has been modified to:**

**"As anthropogenic activities directly and indirectly increase carbon dioxide ($CO_2$) and decrease oxygen ($O_2$) concentrations in the ocean system, it becomes important to understand how different populations of marine animals will respond."**

L72: This sentence is misleading and tends to indicate that with OA all oceans will be undersaturated.
**The sentence has been modified to:**

**"In some regions, as ocean acidification continues, the water becomes undersaturated and corrosive, meaning that, in the absence of compensating biological action, conditions will favor the dissolution of the $CaCO_3$ found in the shells and skeletons of ..."**

L79: Probably the change in saturation state is not the only driver (DIC/proton ratios, pH itself, ..). I would replace this by "modifications of the carbonate chemistry".
**Good point – the sentence has been modified to:**
**"Perturbations of seawater carbonate chemistry can also affect the ability of some calcifying animals…"**

L101: It would be good to compare this value with what is found in the other oceans.
**The sentence has been modified to provide context for the value as:**

**"On top of this natural process, ocean acidification also plays a role: the pH of the upper water column in the North Pacific is decreasing by about 0.002 pH units per year (Byrne et al. 2010; Chu et al. 2016), similar to the global average of 0.0022 pH units per year (Williams et al. 2015). Such a change corresponds to a total $CO_2$, or dissolved inorganic carbon (DIC), increase of 1–2 µmol kg$^{-1}$ yr$^{-1}$ (Peng et al. 2003; Sabine et al. 2008; Sabine and Tanhua 2010; Chu et al. 2016)."**

L146-148: You mention it in the discussion, but it could be interesting to indicate here that they are potentially different species.
**Text has been added as:**

**"The taxonomy of thecosomes has recently begun to be revisited using molecular and paleontological tools (i.e. Hunt et al. 2010; Jennings et al. 2010; Janssen 2012; Maas et al. 2013) and there is growing evidence of cryptic speciation for some pteropod groups (Gasca and Janssen 2014; Burridge et al. 2015). It thus should be noted that these inter-basin comparisons may be of cryptic congeners rather than conspecific populations. Using these organisms, which are presumably adapted to their local conditions, we can test whether species or congeners exhibit a population-specific physiological response to these environmental conditions indicative of different sensitivities."**

L-226: Add "Surface" before "carbonate chemistry".

**The change has been made.**

L340: The effect is probably minor, but pteropod calcification and excretion can change the TA.
**Very true. The sentence has been modified to reflect the uncertainty that is contributed by these processes, but with the assumption that on the timescale of the experiments the influence would be minor:**

**"TA of experimental water was assumed to have been constant over the course of each experiment as water was filtered (0.2 µm) and antibiotic treated (thus microbial activities were kept at minimum). Although pteropod aerobic respiration, excretion and calcification within a respiration chamber could influence TA, it is presumed to have not had a significant influence over the time scales in question."**

L350: Could the difference in TA be due to the bubbling that caused evaporation?
**The bubbling was the same among the different batches of water, and thus it seems likely that the error in TA due to evaporation would have been consistent throughout.**

L456: Increased not decreased?
**Increased. Thanks for the catch!**

The results section contains a large part of methods and discussion. It reads well but I wonder if for clarity the methods and discussion statements should be moved to the corresponding sections.
**In previous versions of the manuscript reviewers found presentation of some of the results confusing. The methods were thus partially repeated in the results section to make sure that the reader understands how the data were collected as it is presented. This was in particular with regards to the carbonate chemistry uncertainty and error. Based on earlier drafts we feel that it is best to retain this material in the same place (results).**

**We have gone through the rest of the results, however, and removed other text that is more discussion based as per the reviewer's recommendations. Specifically, the text that mentions cryptic species has been moved to the introduction and some comments about the distribution of the species to the discussion.**

Burridge AK, Goetze E, Raes N, Huisman J, Peijnenburg KT (2015) Global biogeography and evolution of *Cuvierina* pteropods. BMC evolutionary biology 15 doi 10.1186/s12862-015-0310-8

Byrne RH, Mecking S, Feely RA, Liu X (2010) Direct observations of basin-wide acidification of the North Pacific Ocean. Geophys Res Lett 37: L02601

Chu SN, Wang ZA, Doney SC, Lawson GL, Hoering KA (2016) Changes in anthropogenic carbon storage in the Northeast Pacific in the last decade. Journal of Geophysical Research: Oceans 121 doi 10.1002/2016JC011775

Gasca R, Janssen AW (2014) Taxonomic review, molecular data and key to the species of Creseidae from the Atlantic Ocean. Journal of Molluscan Studies 80: 35-42

Hunt B, Strugnell J, Bednarsek N, Linse K, Nelson RJ, Pakhomov E, Seibel B, Steinke D, Würzberg L (2010) Poles Apart: The "Bipolar" Pteropod Species *Limacina helicina* Is Genetically Distinct Between the Arctic and Antarctic Oceans. PLoS ONE 5: e9835

Janssen AW (2012) Late Quaternary to Recent holoplanktonic Mollusca (Gastropoda) from bottom samples of the eastern Mediterranean Sea: systematics, morphology. Bollettino Malacologico 48: 1-105

Jennings RM, Bucklin A, Ossenbrügger H, Hopcroft RR (2010) Species diversity of planktonic gastropods (Pteropoda and Heteropoda) from six ocean regions based on DNA barcode analysis. Deep Sea Research Part II: Topical Studies in Oceanography 57: 2199-2210

Maas AE, Blanco-Bercial L, Lawson GL (2013) Reexamination of the species assignment of Diacavolinia pteropods using DNA barcoding. PLoS ONE 8: e53889 doi doi:10.1371/journal.pone.0053889

Peng T-H, Wanninkhof R, Feely RA (2003) Increase of anthropogenic $CO_2$ in the Pacific Ocean over the last two decades. Deep Sea Research Part II: Topical Studies in Oceanography 50: 3065-3082

Sabine CL, Feely RA, Millero FJ, Dickson AG, Langdon C, Mecking S, Greeley D (2008) Decadal changes in Pacific carbon. J Geophys Res Oceans 113: -

Sabine CL, Tanhua T (2010) Estimation of anthropogenic $CO_2$ inventories in the ocean. Annu Rev Mar Sci 2: 175-198

Williams NL, Feely RA, Sabine CL, Dickson AG, Swift JH, Talley LD, Russell JL (2015) Quantifying anthropogenic carbon inventory changes in the Pacific sector of the Southern Ocean. Marine Chemistry 174: 147-160

---

## Author Response (AR2)

**Associate Editor Decision: Publish subject to minor revisions (Editor review)** (14 Oct 2016) by Dr. Jean-Pierre Gattuso
Comments to the Author:
Dear Author,

Thank you for submitting a revised version of your manuscript submitted to Biogeosciences, which can be accepted for publication after minor revision. When submitting the revised version, please let me know which of the changes were not implemented, if any, and why. This will speed up final acceptance.

I look forward to seeing this paper published and thank you for considering Biogeosciences to publish these very interesting results.

Best regards,
Jean-Pierre Gattuso
BG editor

**Dear Dr. Gattuso,**

**Thank you for your assistance moving our manuscript closer to publication. Below are attached your recommendations, our changes and/or an explanation of why the change was not implemented.**

**All the best,**
**~Amy Maas**
* * *
- The referees asked to report the size range. In your reply, you say that the size range is already reported, as mass. Mass is not size! I appreciate that some sizes are not available but you could provide the measurements you have (length or diameter, depending on the shape of the species considered).

**Unfortunately, measurements of size were not made at the time of the respiration experiments, in the interest of minimizing any effect of handling of animals. Following the experiments, the specimens were immediately frozen for later molecular work; some of these individuals have already been used for analyses of gene expression, and we are highly reluctant to thaw any of those that remain as this would make them unusable for future molecular work. Thus, this change was not implemented.**

- I wonder how the percent saturation was calculated. For example, in the abstract you mention 130 µmol/kg is 10%. Using the standard equation of Garcia & Gordon (1992, L&O), the O2 saturation at S=35, T=5, depth=200 m, is 307.892 µmol/kg. Hence, 130 µmol/kg would correspond to 42% of the saturation value.

**We had an internal discussion about this percent saturation attribution as well. As you know, water that is fully saturated with oxygen only has 21%, the rest being made up of nitrogen. To order gas of the concentration that is appropriate for the Pacific we thus ordered 10% oxygen. This is ~48% of what would be oxygen saturation (similar to your calculation, minus the fact that we did not take into**

**account the 200 m depth, just the salinity and temperature as the experiments were run at sea level). As this appears to be confusing, for clarity, we have gone back through and re-expressed the % as relative to oxygen saturation rather than total gas composition for both 21% (100% saturated) and 10% (48% saturated).**

- Mention in section 2.2 that pH was measured on the total scale

**The information of pH in total scale has already been provided on line 200 in section 2.2. We have added the abbreviation to draw attention to the scale.**

- 224 and elsewhere: always use the subscript "T" when an absolute pH value is given, including in the heading of Table 2. The heading of that table should also provide the unit for temperature.

**The changes have been made.**

- aAbreviate "hours by "h"

**The change has been made**

- 272: ranged between 15 and 50 ml ... and 8 to 20 ml (similar changes needed line 323)

**The changes have been made (as well as in line 273).**

- 337-338: the symbol for mole is "mol", not "M". But in this sentence mole could be spelled out.

**The "M"s have been changed to moles.**

- 337: indicate for which species Mayzaud reported this respiratory quotient of 0.8.

**The information has been added**

- Please list citations chronologically throughout the manuscript.

**The change has been made**

- References need to be formatted as described in the instructions to authors.

**The references have been carefully checked to meet the formatting requirements of Biogeosciences.**

- Biogeosciences strongly promotes the full availability of the data sets reported in the papers that it publishes in order to facilitate future data comparison and compilation as well as meta-analysis. This can be achieved by uploading the data sets in an existing database and providing the link(s) in the paper. Alternatively, the data sets can be published, for free, alongside the paper as supplementary information. The ascii (or text) format is preferred for data and any format can be handled for movies, animations etc…

**The respiration data is available online via the DOI provided in the text. The carbonate chemistry measurements and calculations associated with the experiments have now been included as supplementary data. The environmental chemistry measurements for the Pacific are available in BCO-DMO (Chu et al 2016) and the Atlantic data is in preparation for paper submission and will be available in BCO-DMO after paper acceptance.**

[revised manuscript text omitted]